# Highly efficient, heat dissipating, stretchable organic light-emitting diodes based on a MoO₃/Au/MoO₃ electrode with encapsulation

Dae Keun Choi[1,2,3,12], Dong Hyun Kim[1,2,3,12], Chang Min Lee[1,2,3,12], Hassan Hafeez [1,2 ✉], Subrata Sarker [1,2,3], Jun Su Yang[1], Hyung Ju Chae[1,2,3], Geon-Woo Jeong[1,2,3], Dong Hyun Choi[1,2,3], Tae Wook Kim[1,2,3], Seunghyup Yoo[4], Jinouk Song[4], Boo Soo Ma[5], Taek-Soo Kim[5], Chul Hoon Kim[6], Hyun Jae Lee[6], Jae Woo Lee [7], Donghyun Kim [7], Tae-Sung Bae[8], Seung Min Yu[8], Yong-Cheol Kang[9], Juyun Park[9], Kyoung-Ho Kim[10], Muhammad Sujak [10], Myungkwan Song [11], Chang-Su Kim [11 ✉] & Seung Yoon Ryu [1,2,3 ✉]

Stretchable organic light-emitting diodes are ubiquitous in the rapidly developing wearable display technology. However, low efficiency and poor mechanical stability inhibit their commercial applications owing to the restrictions generated by strain. Here, we demonstrate the exceptional performance of a transparent (molybdenum-trioxide/gold/molybdenum-trioxide) electrode for buckled, twistable, and geometrically stretchable organic light-emitting diodes under 2-dimensional random area strain with invariant color coordinates. The devices are fabricated on a thin optical-adhesive/elastomer with a small mechanical bending strain and water-proofed by optical-adhesive encapsulation in a sandwiched structure. The heat dissipation mechanism of the thin optical-adhesive substrate, thin elastomer-based devices or silicon dioxide nanoparticles reduces triplet-triplet annihilation, providing consistent performance at high exciton density, compared with thick elastomer and a glass substrate. The performance is enhanced by the nanoparticles in the optical-adhesive for light out-coupling and improved heat dissipation. A high current efficiency of ~82.4 cd/A and an external quantum efficiency of ~22.3% are achieved with minimum efficiency roll-off.

[1] Division of Display and Semiconductor Physics, Display Convergence, College of Science and Technology, Korea University Sejong Campus, Sejong City, Republic of Korea. [2] Department of Applied Physics, Korea University Sejong Campus, Sejong City, Republic of Korea. [3] E-ICT–Culture-Sports Convergence Track, Korea University Sejong Campus, Sejong City, Republic of Korea. [4] School of Electrical Engineering, Korea Advanced Institute of Science and Technology (KAIST), Daejeon, Republic of Korea. [5] Department of Mechanical Engineering, Korea Advanced Institute of Science and Technology (KAIST), Daejeon, Republic of Korea. [6] Department of Advanced Materials Chemistry, College of Science and Technology, Korea University Sejong Campus, Sejong City, Republic of Korea. [7] Interdisciplinary Graduate Program for Artificial Intelligence Smart Convergence Technology, Korea University, Sejong, Republic of Korea. [8] Jeonju Center, Korea Basic Science Institute (KBSI), Analysis & Researcher Division, Jeollabuk-do, Republic of Korea. [9] Department of Chemistry, Pukyong National University 45 Yongso-Ro, Nam-gu, Busan, Republic of Korea. [10] Department of Physics, Chungbuk National University, Cheongju, Republic of Korea. [11] Surface Materials Division, Korea Institute of Materials Science (KIMS), Changwon, Republic of Korea. [12] These authors contributed equally: Dae Keun Choi, Dong Hyun Kim, Chang Min Lee. ✉email: hassaniskt@hotmail.com; cskim1025@kims.re.kr; justie74@korea.ac.kr

The challenges for organic light-emitting diodes (OLEDs)[1] with a perfect form factor have led to research into advanced manufacturing processes. Incorporation of OLEDs into wearable[2], epidermal[3], and stretchable electronics[4,5], with color stability, and high brightness under intensely strained bending/stretching has been of major interest. This interest has led to various studies on electrodes as a replacement of poly(3,4-ethylenedioxythiophene):poly(styrene sulfonate) (PEDOT:PSS)[6], and indium tin oxide (ITO)[7–9], which included a graphene-based electrode[10], a highly transparent metal-grid[11] and polymer with silver nanowires (AgNWs)[12,13]. Even though these approaches provide considerable enhancements for flexible/stretchable electronics, there is still scope for further improvement in terms of device stability and efficiency[6,8,9,12–14]. Transparent electrode based on dielectric/metal/dielectric and an oxide-metal-oxide (OMO), where a thin metal layer is sandwiched between two dielectric or oxide layers, is similarly a promising candidate. That can be tuned to produce a highly conductive and transparent multilayer electrode exploiting the conductivity of the metal, the optical interference within the multilayers, and the surface plasmonic effect at the metal/oxide interface[15–19]. In terms of flexibility and stretchability, a thin and ductile metal layer has the role of resisting fracture since it acts as the crack stopper or retardation on an elastomeric substrate[19]. Therefore, a thin OMO electrode could be proposed for stretchable electronics because of its better hole injection and optical transmittance, despite the thick oxide material being known for brittleness under high strains[20]. Among the OMOs, molybdenum trioxide ($MoO_3$)/gold (Au)/$MoO_3$ (MAM) has been found to be one of the promising candidates for the highly conductive and transparent electrodes due to the deep work function of $MoO_3$ and Au for hole injection, including the ductile property for stretchability, even though the $MoO_3$ dissolves in water and Au is expensive[15,21,22].

Moreover, a flexible or stretchable substrate is important. Polydimethylsiloxane (PDMS) has been widely used due to its high stretchability because of a low Young's modulus (YM) of ~2 MPa[23]. However, it has demonstrated poor adhesion due to hydrophobicity[24]. Polyurethane is another option due to its low YM of ~280 MPa[25]. However, the hygroscopicity of the material made it unreliable, when used for stretchable polymer LEDs[25]. An ultra-violet (UV) curable polymer, Norland Optical Adhesive 63 (NOA63), has proven to be an excellent candidate due to its mechanical stability and good adhesion with the electrode because of its hydrophilicity[26]. A high YM of ~1.5 GPa[8], can cause poor stretchability, but if utilized as a thin substrate with an elastomer, NOA63 attached to 3 M elastomer can be suitable for flexible or stretchable electronics applications[8,27–29]. However, heat generation during operation may cause degradation of an OLEDs[30,31], such as exciton quenching from triplet–triplet annihilation (TTA) and triplet-polaron annihilation (TPA) at high exciton densities in thick substrate[32–34]. There are two strategies to dissipate the generated heat from the OLEDs; one is the "heat sink" that requires an additional thermally conductive layer in the device and the other one is the "heat dissipation" through the thin substrate and encapsulation layer that does not require an extra conductive layer[35]. Especially, in OLEDs, a short heat transfer pathway can further facilitate heat dissipation without using a heat sink[30]. This pathway can be provided by the incorporation of metal oxide nanoparticles (NPs) in the thin substrate and can enhance the heat dissipation in optoelectronic devices due to their high, thermal conductivity and porous scaffold design[35].

Various types of stretchable OLEDs including GSOLEDs (geometrically stretchable OLEDs, transferred to the pre-strained substrate, buckled after strain release)[27–29,36] and ISOLEDs (intrinsically stretchable OLEDs directly stretched)[5] were previously introduced. Stretchable active-matrix OLEDs[37], ultra-thin polymer LEDs[36], stretchable inorganic LEDs[38], and OLEDs with thermally pre-strained PDMS[39] have been demonstrated. However, there are some concerns about the deviations in the electroluminescence (EL) and color coordinates, by the semi-transparent electrode (Ag 18 nm, aluminum (Al) 20 nm) due to the strong microcavity effect[27–29,39]. Such layouts demonstrated a shifted EL between the buckled (non-stretched) and planar (stretched) modes, while stretchability of the GSOLEDs were also limited to 50% or less[28,39]. Although the passivation for flexible and stretchable OLEDs have been demonstrated previously[40–42], various approaches of encapsulation for water-proofing, and mechanical analysis of the strain have not been discussed.

This research realized twistable GSOLEDs utilizing the superior optoelectronic characteristics of MAM electrodes. A thin NOA63 film attached to the pre-strained thick 3 M elastomer with a low YM of ~1.4 MPa and thin elastomer induced a small mechanical bending strain; thus, the OMO interface including the organic layers was prevented from a mechanical failure[8,27–29]. The MAM-based twistable GSOLEDs with encapsulation showed high external quantum efficiency (EQE) and color stability at various strains and angles without efficiency roll-off until 100% strain by heat dissipation, which made the device unique compared to the other GSOLEDs (Supplementary Table S1)[28,39]. To further enhance the efficiency, silicon dioxide ($SiO_2$) NPs[43], which enable the extraction of the confined light, were added to the NOA63 film achieving around 10% improvement in the out-coupling efficiency.

## Results

**GSOLED device configuration with MAM electrode.** Figure 1a shows a schematic illustration of the device structure, which is described in detail in the Methods section. The pristine NOA63 without NPs and the modified NOA63 with $SiO_2$ 470 nm sized NPs substrates were prepared. The thickness of the NOA63 (~16.3 µm) with NPs and without NPs (~9.2 µm) was measured using ultra-high-resolution field emission scanning electron microscopy (UHR FE-SEM). The higher thickness of NOA63 with NPs could be ascribed to the surface modification and agglomeration of NPs by the UV treatment[35,44,45]. The GSOLEDs were fabricated on both substrates using the same device layout with a MAM electrode[15–17]. The bandgap alignment of the layers is shown in Fig. 1b. The MAM demonstrated a superior φ (5.25 eV) as compared to the Ag electrode (4.81 eV), due to the deeper conduction band of the $MoO_3$, analyzed using ultraviolet photoelectron spectroscopy (UPS) (Supplementary Fig. 1). Transition metal oxides are utilized to reduce the hole injection barriers, hence, the sandwiched $MoO_3$ with Au is beneficial for better device performance[46,47]. Moreover, the MAM thicknesses were optimized using thin-film optic theory and sheet resistance analysis[15–17]. $MoO_3$(15 nm)/Au(14 nm)/$MoO_3$(5 nm) demonstrated a fair balance between high transmittance of 70–80% and low sheet resistance of ~20 Ω/sq, the best figure of merit[48,49] value ($2.96 \times 10^{-3} \, \Omega^{-1}$) (Supplementary Fig. 1). The prepared MAM electrode showed a stable performance during the stretching cyclic test (Supplementary Fig. 2 and Supplementary Movie 1). The device stability was also reinforced by using thin NOA63 for encapsulation, which fabricated a sandwiched structure with thin NOA63, induced a small bending strain, and adjusted the mechanical neutral plane (MNP) at the high YM layers[8,27–29]. Further, the encapsulation of NOA63 with 3 M tape, silicon nitride ($SiN_x$) (200 nm) thin film and NOA63 side passivation presented a moderate water vapor transmission rate (WVTR) ($3.6 \times 10^{-2} \, \mathrm{g \, m^{-2} \, day^{-1}}$, Supplementary Fig. 3) as compared with that of the recently introduced 4–6 layers of

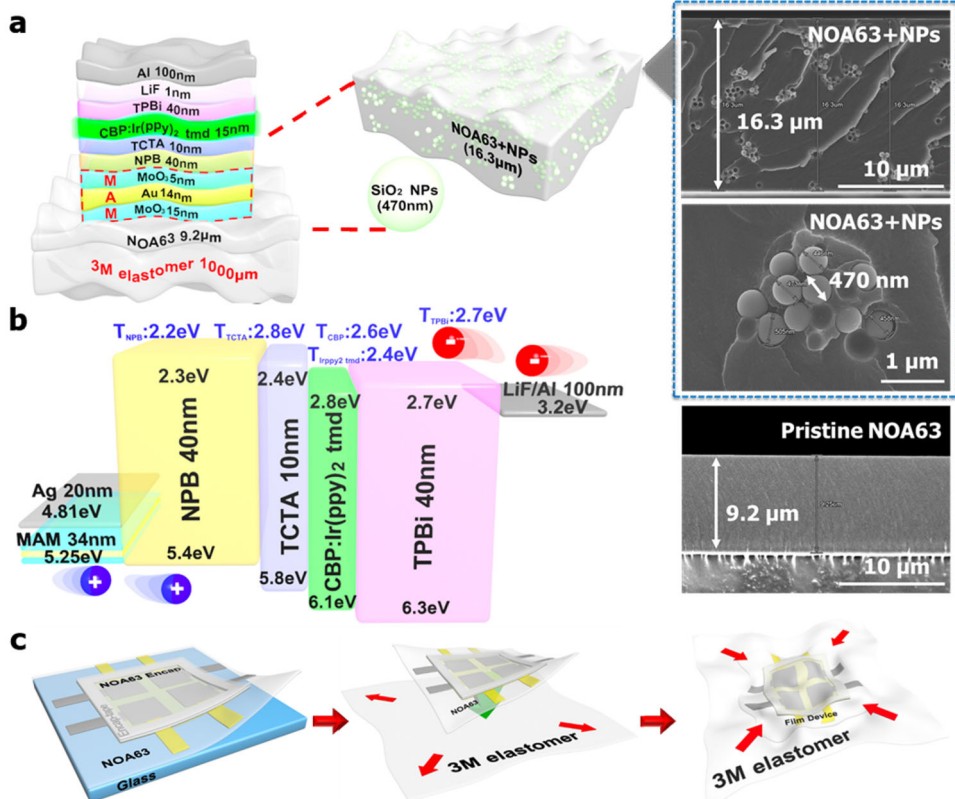

**Fig. 1 Schematic illustration of the device layout and fabrication process along with the bandgap alignment. a** The schematic illustration of the geometrically stretchable organic light-emitting diodes (GSOLEDs) device structure along with thickness of the layers. Extended figures demonstrate the thickness of the Norland Optical Adhesive 63 (NOA63) with and without silicon dioxide nanoparticles (SiO$_2$ NPs), size analysis of the NPs using ultra-high-resolution field emission scanning electron microscopy (UHR FE-SEM). **b** Bandgap diagram of the layers and their triplet energy values with molybdenum trioxide (MoO$_3$)/gold (Au)/MoO$_3$ (MAM) electrode. **c** Schematic illustration of the steps involved in the GSOLED device fabrication including NOA63 thin-film encapsulation and peeling off film device, 3 M elastomer pre-strain and attach film device, release, and stretch GSOLEDs.

graphene-encapsulated OLEDs ($1.78 \times 10^{-2}$ g m$^{-2}$ day$^{-1}$)[40,41]. Nevertheless, there is a trade-off between encapsulation quality and stretchability. Therefore, further study is needed to implement edge encapsulation with a stretchable sealant for the industrial requirement and practical wearable gadgets based on the GS- or ISOLEDs.

The encapsulated devices were then peeled from the glass substrate and transferred to a thick mechanically pre-strained, very high bond (VHB) 3 M elastomer (1000 μm) (Figs. 1c, 2a, Supplementary Fig. 4 and Supplementary Movie 2). The process was similar to kinetic transfer printing but without a bulky transfer stamp on top[50]. This was because the peel-off was assisted with the hydrophobic detergent (detailed description in the Methods section) used for the surface treatment. After transfer, the two-dimensional (2D) mechanical strain on the 3 M elastomer was released and the 2D shrinkage of the elastomeric tape resulted in random wavy buckles in the device[51]. GSOLED device performance images and stability at various 2D stretch percentages (0–100%) were analyzed at the 2D stretchable mode. The devices were also 1D twisted at various angles (0–180°, Supplementary Movies 3 and 4), and performance stability was tested in the 2D flexible mode by attaching the device to a golf ball (standard diameter = 42.67 mm, Supplementary Movie 5). The water immersion test on the NOA63 encapsulation, on top of the device, was conducted. This sandwiched structure demonstrated significant stability in liquid environments (Fig. 2a–c and Supplementary Movie 6). The fabricated wavy buckles were observed by optical microscope (OM) and confocal microscope (CM) on the NOA63, with and without NPs. We have also

measured the pixel area and non-pixel area with CM for verifying buckling on the different parts of the devices (Supplementary Figs. 5 and 6). The buckling of thin elastomer showed smaller buckling compared to thick 3 M elastomer. In addition, we have defined the strain based on light-emitting area (pixel area) with both types of elastomers (Supplementary Fig. 7).

**Device performance with and without SiO$_2$ NPs and heat dissipation mechanism.** The device performances have been compared 'GS devices' between thick 3 M elastomer and thin elastomer. This differentiated between 'stretchable & non-heat dissipating devices' and 'stretchable & heat-dissipating devices'[30–34] (Fig. 3a, d). The GS devices with 3 M elastomer could not dissipate heat efficiently, while the GS devices with thin elastomer (constraint by the pre-strain value ~30%) dissipated heat efficiently. As the heat generated by triplet exciton annihilation is accumulated and not easily dissipated through the thick elastomer, there was a bit of efficiency roll-off issue, which was unstable even at high exciton densities over 10,000 nits. Moreover, SiO$_2$ NP clusters of ~470 nm were optimized and incorporated into the NOA63 substrate to enhance the EQE by light scattering[43] (Fig. 3b, c). In addition, we have replaced the thick 3 M elastomer with another thin elastomer (100 μm, DOW Corning HTV processed by Apple Silicone, Korea) and the detailed device physics is discussed later.

The device performance of the GSOLEDs at various stretch percentages (0–100%) was demonstrated and the results are shown in Table 1. In GSOLEDs, a phosphorescent horizontal emitter (bis(2-phenylpyridine)iridium(III) (2,2,6,6-tetramethylheptane-3,5-diketonate) [Ir(ppy)$_2$tmd] resulted in higher out-coupling efficiency

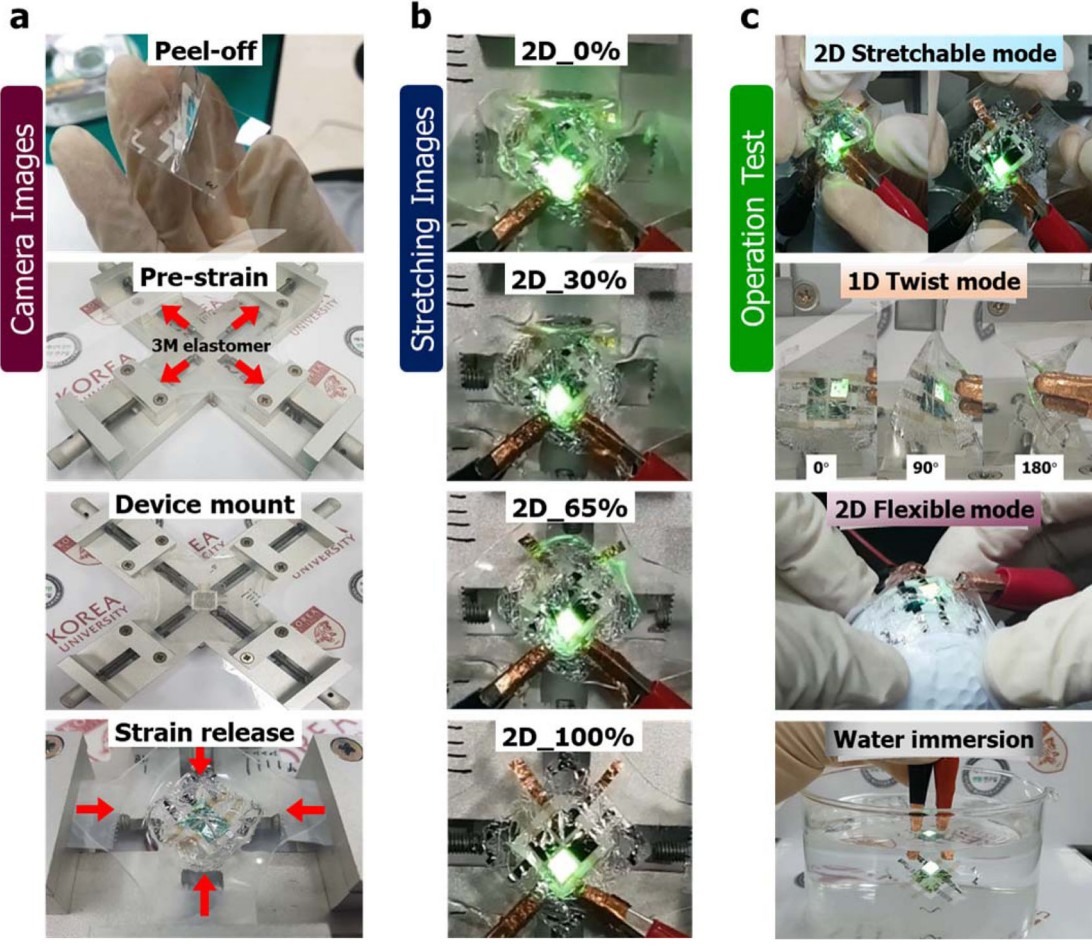

**Fig. 2 Camera images of the device transfer, performance, and stability tests. a** Camera images demonstrating the fabrication process of device transfer from glass substrate to a mechanically pre-strained flexible/stretchable 3 M elastomer (clamped in the jig). **b** GSOLED device performance images at various two-dimensional (2D) stretch percentages (0–100%). **c** The stability and performance of GSOLED devices analyzed at 2D stretchable mode. The NPs were not incorporated in these devices. Also, the devices were one-dimensional (1D) twisted at various angles (0–180°) and tested in the 2D flexible mode by attaching the device to a golf ball (standard diameter = 42.67 mm). The water immersion test was conducted. The sandwiched device structure demonstrated significant stability in liquid environments due to NOA63 encapsulation.

than isotropic orientation of dipoles in OLEDs[52,53]. The current efficiency (CE), power efficiency (PE), and EQE are 75.4 cd A$^{-1}$, 47.4 lm W$^{-1}$, 20.6% without NPs, and 82.4 cd A$^{-1}$, 57.5 lm W$^{-1}$, 22.3% with NPs, respectively (Table 1)[52,53]. The device performance with the NOA63 substrate with SiO$_2$ NPs was not affected until 100% stretching due to the proposed strategy for GSOLEDs. The efficiency enhancement by SiO$_2$ NPs was further evaluated by reflectance, haze analysis, and scattering (Supplementary Fig. 8). A decrease in reflectance was observed for the SiO$_2$ NP-incorporated NOA63 as compared to pristine NOA63 without NPs, which was offset by the high haze, indicating a forward Mie scattering effect[54,55] by the NPs. However, the NPs-incorporated devices were fabricated in the buckled or wavy modes with larger buckle dimensions (Supplementary Fig. 6). We speculate that the increased YM with the addition of 470 nm SiO$_2$ NPs could have resisted the mechanical buckling process.

To validate stretchability and heat dissipation properties, additional devices have been fabricated using a thin elastomer (100 μm) at the different strain conditions (0, 10, 20, and 30%) as shown in Fig. 3d. The heat dissipation function of the devices based on the thin elastomer was much efficient than those based on thick 3 M elastomer, but less efficient than the thin NOA63 (9.2 μm) layers (Supplementary Fig. 9) especially at low exciton density (below 10,000 nits) due to the thickness of the elastomer.

However, they performed similarly over 10,000 nits (Fig. 3e). The CE with thin elastomer was a bit lower than that of only thin NOA63 substrate due to thin elastomer thickness (100 μm), which means that a small quantity of heat still might not have dissipated from thin elastomer. Moreover, the stretchability constraint of thin elastomer (~30%) was much lower than that of 3 M elastomer (~100%). We have also tried to fabricate stretchable and heat-dissipating devices with even thinner elastomer (<100 μm); however, it was hard to be fabricated due to the handling difficulties of being quick degradation over 30% strain.

As suggested in Fig. 2, the twist mode and the device lifetime (LT) under strain conditions have been conducted as shown in Fig. 4a–f. The device performance at various twisting angles (0°, 90°, 180°) has been measured and the GSOLEDs were unaffected by the twisting. And, the twisting cyclic test did not show any change in the device performance up to 100 cycles. The LT50 at the different strain conditions (0, 30, 65, 100%) was also analyzed with 3 M elastomer and the tendency remained the same despite the strain. It means that geometrical stretching does not affect the device operation due to mechanical consideration. Moreover, the mechanical stress on various applied strains of the GSOLEDs due to the fabricated wavy buckles was analyzed using FE simulation (Fig. 4g and Supplementary Fig. 10)[9,56]. The randomly

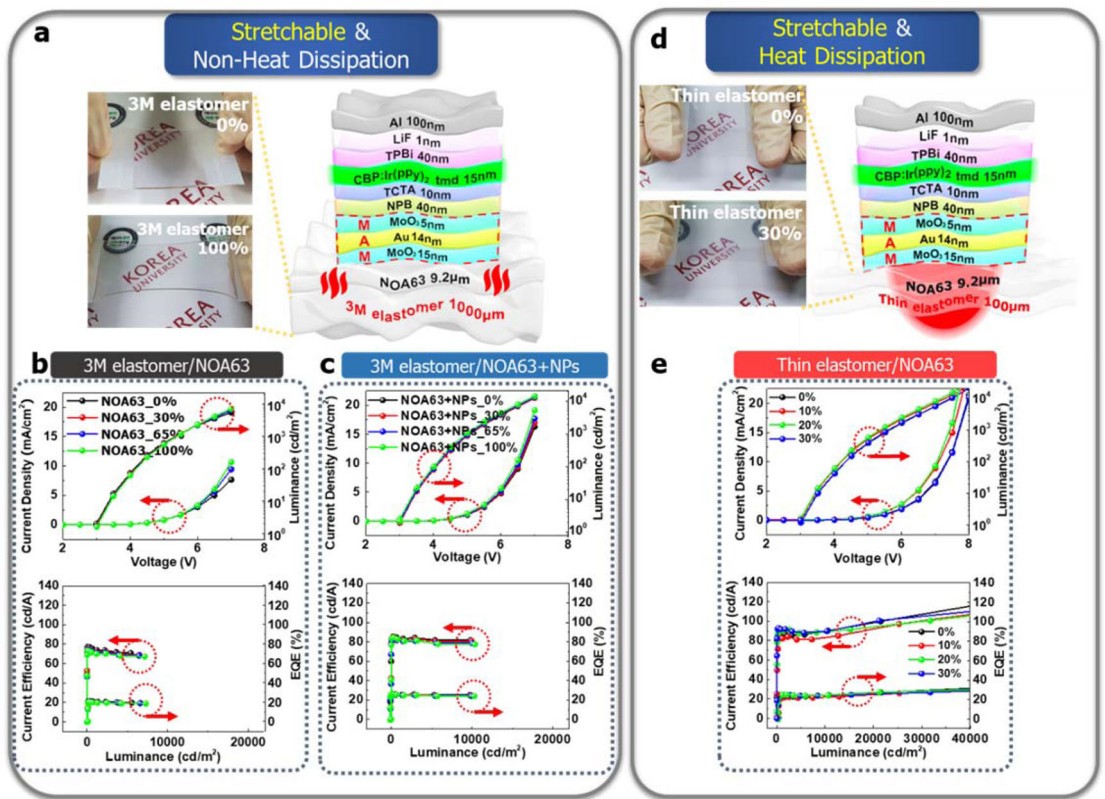

**Fig. 3 Device performance of the stretchable and heat-dissipating devices based on MoO₃/Au/MoO₃ (MAM) with and without SiO₂ NPs. a, d** The schematic illustration of GSOLEDs and 'Stretchable & Non-Heat dissipating devices' with the 3 M elastomer, and 'Stretchable & Heat dissipating devices' with the thin elastomer, respectively. Insets of a, d, show camera image of the 3 M elastomer (0 and 100% strain) and the thin elastomer (0 and 30% strain). **b** Current density-voltage-luminance, current efficiency, and quantum efficiency analysis at various applied 2D strains with 3 M elastomer and Norland Optical Adhesive 63 (NOA63) film. **c** With 3 M elastomer and NOA63 + silicon dioxide nanoparticles (SiO₂ NPs). **e** with thin elastomer.

**Table 1 Device performance (current efficiency, power efficiency, and external quantum efficiency) of the GSOLED devices on thin Norland Optical Adhesive 63 (NOA63)/3 M elastomer substrate, with MAM electrode using isotropic and horizontal emitter at low luminance (500 cd/m².**

|  | 2D area strain (%) | Current efficiency (cd/A) | Power efficiency (lm/W) | External quantum efficiency (%) | CIE Color coordinates (x, y) |
|---|---|---|---|---|---|
| NOA 63 | 0 | 75.4 | 47.4 | 20.6 | (0.340, 0.621) |
|  | 30 | 73.4 | 46.1 | 20.1 | (0.340, 0.621) |
|  | 65 | 72.8 | 45.7 | 19.9 | (0.340, 0.621) |
|  | 100 | 71.7 | 45.1 | 19.6 | (0.340, 0.621) |
| NOA63 + SiO₂ NPs | 0 | 82.4 | 57.5 | 22.3 | (0.333, 0.625) |
|  | 30 | 81.2 | 56.7 | 22.0 | (0.334, 0.625) |
|  | 65 | 82.0 | 57.2 | 22.2 | (0.334, 0.625) |
|  | 100 | 84.4 | 58.9 | 23.9 | (0.334, 0.625) |

Color coordinates are presented in compliance with International Commission on Illumination (CIE-1931).

wrinkled GSOLED was modeled using eigenvalue buckling analysis. As the stretching strain increased by 0, 30, 65, and 100%, the stresses of the OLED layers were least affected by the overall deformation in the entire device including 3 M elastomer. At a 100% stretching strain of 3 M elastomer, the overall strain level of the OLED device was ~5%, which is considerably lower than that of the 3 M elastomer. The low stress in the pixel part was due to the wrinkled structure, formed intensively in the pixels through the pre-strain method, that could effectively release the mechanical strain of the device during the application of the tensile stress. Similar to the simulation results, the high

mechanical reliability of the wrinkled structure in the device was also confirmed by various stability tests. Despite of the strain value of the device (<9%) in simulation, the actual device was not damaged in the stability tests since crack initiation and propagation was prevented by the adhesion of the sticky elastomer substrate[57–59]. The simulation result suggests that the geometrically wrinkled structure is suitable to fabricate stretchable OLEDs. Also, the experimental results on cyclic and twist tests support the simulation result. As the OLED devices were encapsulated in a sandwiched structure having a thin NOA63 (9.2 μm) at the top and bottom, the analysis indicates that the

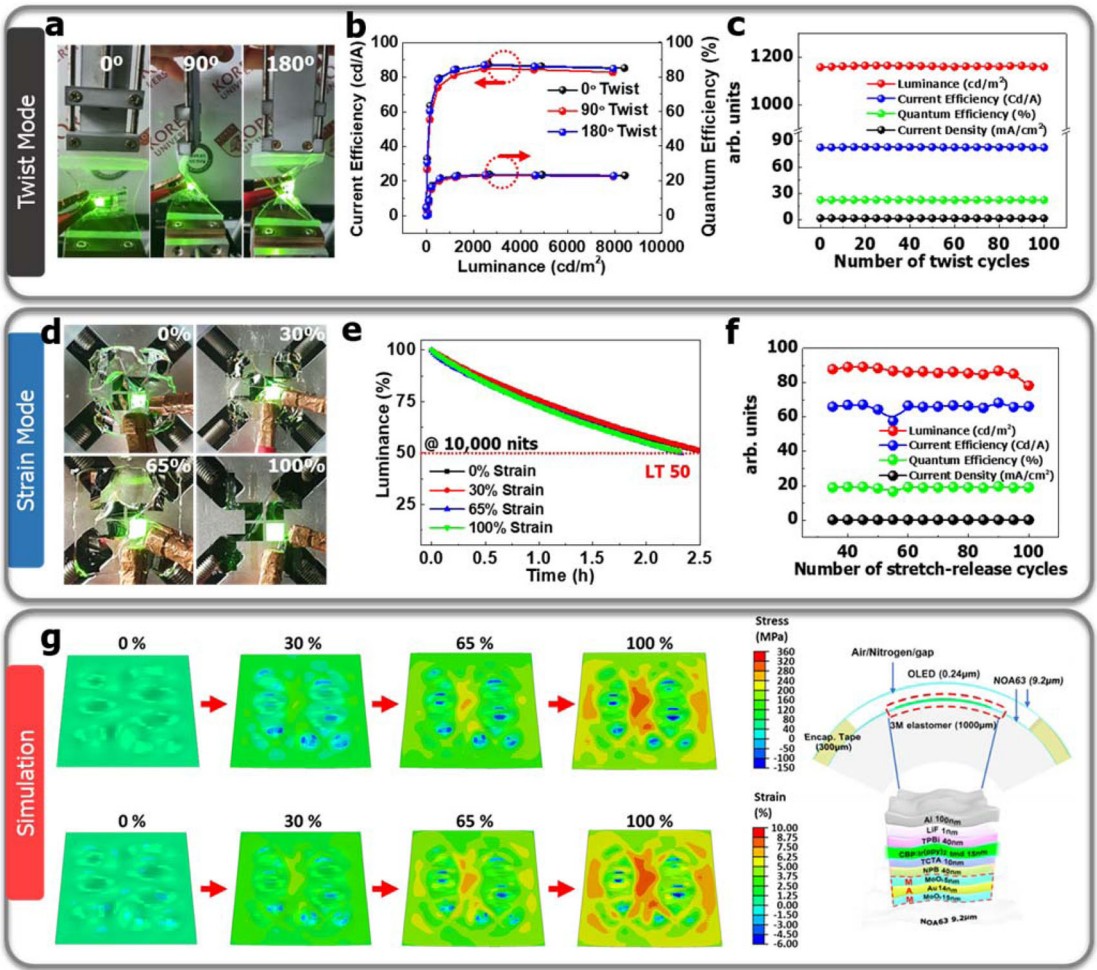

**Fig. 4 Device performance of twist mode and strain mode and mechanical simulation of the GSOLED based on 3 M elastomer. a** Camera images of the device at different twist angles (0°, 90°, and 180°). **b** Current efficiency and quantum efficiency at different twist angles. Inset of **b** shows EL intensity at different twisting angles. **c** Device performance including luminance (red), current efficiency (blue), quantum efficiency (green), and current density (black) against 100 times twisting cycles. **d**, Camera images of the device on strain condition (0, 30, 65, and 100%). **e** Lifetime 50% (LT 50) of the devices at various 2D stretch percentages (0–100%). **f** The device performance stability of the cyclic test was further evaluated using a stretch-release cyclic test where the devices were tested for up to 100 cycles at an applied strain of 30%, demonstrating high mechanical robustness. **g** Mechanical simulation to demonstrate the strain induced on the GSOLEDs by the wavy buckle formation and schematic illustration showing hollow encapsulation of NOA63 on top of the device structure.

small bending strain and the MNP were suitably adjusted to minimize the damage to the efficient layers[8,27–29] (Supplementary Fig. 11). Thus, the theoretical simulation shows that the device structure acquired to fabricate GSOLED devices is mechanically sustainable and efficient.

Figure 5a, b suggests the following heat dissipation mechanism. The heat conductivity is related to the thickness normalization, which was 0.2 and 1.05 W m$^{-1}$ K$^{-1}$ for thin NOA63 and thick glass substrates, respectively. This indicates that the glass substrate itself intrinsically transfers more heat than a NOA63 film[30,31]. However, the heat conductance (along the direction normal to the substrate) is dependent on the relative substrate thickness, which for thin NOA63 and the thick glass substrate are 0.543 and 0.0375 [W K$^{-1}$]. This means that a thin NOA63 substrate would transfer heat more efficiently to ambient air than a thick glass substrate[30,31] (Supplementary Fig. 12). This modeling is described in the schematic illustration, and the real hot spot size of the thick glass substrate device is wider than that of the thin NOA63 substrate in infra-red (IR)-camera images (Fig. 5b). Thermal simulation, temperature observations from the real IR-camera images, and the consequential device

performances for thickness-dependent NOA63 substrate (9.2, 70, 210 µm) are provided (Supplementary Figs. 9 and 13). The temperature drop rate of the thin NOA63 substrate surface was faster than that of thick NOA63 or glass substrate after the device was turned off, which corresponds to the conductance difference between the thin NOA63 substrate and thick glass (Fig. 5b, c and Supplementary Figs. 12 and 13). This heat accumulation in thick substrates and heat dissipation in thin substrates induced different current injection, which was confirmed by pulsed I–V ($W_{pulse} = 10$ µs, $\tau_{rise} = 1$ ns, $\tau_{fall} = 1$ ns) and direct-current (DC) I–V characteristics of the devices. Pulsed measurement within a short term did not induce heat generation, while DC operation triggered heat generation within the device. The current injection in the thick-substrate devices (glass and NOA63 with 210 µm) was delayed, compared with that of the thin substrate device (9.2 µm) because heat generation hinders the current injection into devices from the ideal diode equation (Fig. 5d and Supplementary Fig. 14)[60]. The generated and accumulated heat in thick-substrate devices provided thermally activated energy to triplet excitons, which boosted longer exciton lifetimes, TTA, and TPA, finally degrading the device efficiency at high exciton density[30–34].

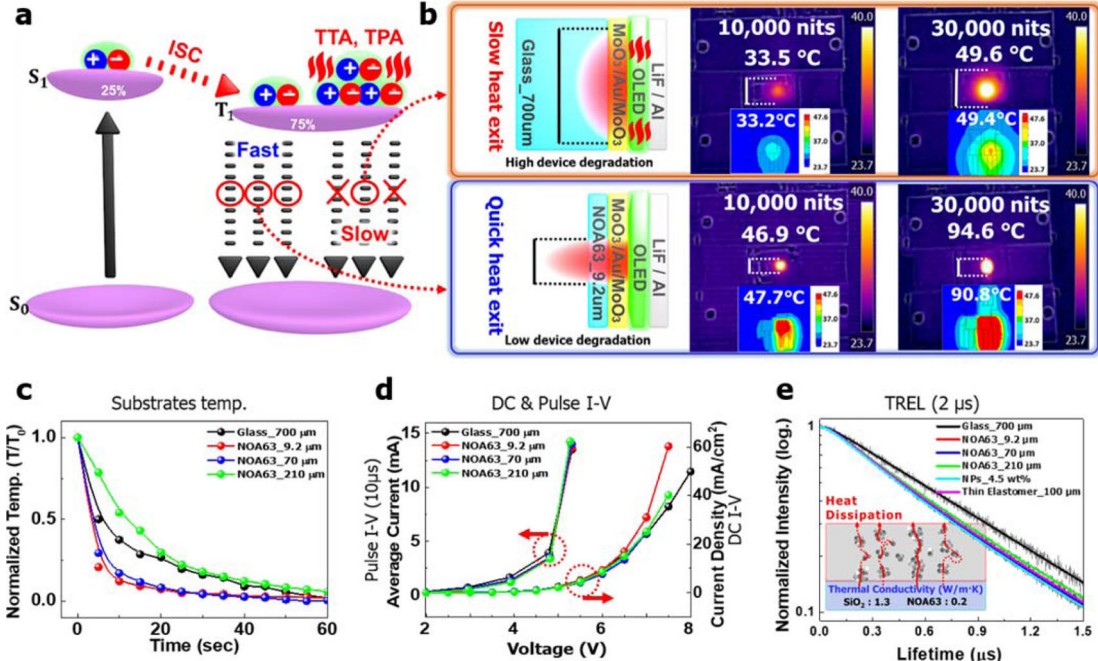

**Fig. 5 Device performance of the MoO$_3$/Au/MoO$_3$ (MAM) based GSOLED devices by heat dissipation mechanism. a** The schematic illustration of the heat dissipation mechanism based on the triplet–triplet annihilation (TTA) and triplet-polaron annihilation (TPA) process **b** The schematic illustration, infrared (IR)-camera, and thermal simulation of heat accumulated and dissipated tendency in thin Norland Optical Adhesive 63 (NOA63) and thick glass substrate, which was confirmed by the temperature observance of the various substrate surfaces **c** The temperature drop rate curves with various substrates after the device is turned off. **d** The current density curve for pulsed I–V (W$_{pulse}$ = 10 μs, duty cycle = 10%) and direct-current (DC) I–V characteristics of devices. **e** The time-resolved electroluminescence (TREL) decay profiles (thin gray lines) and their multi-exponential fitting results (thick solid lines), which reveal that the exciton lifetimes of the EL devices can be gradually disturbed by adjusting their substrate thickness. Introducing nanoparticles (NPs) and a thin elastomer to the device also showed a similar effect. The inset of e describes the schematic illustration of metal oxide NPs clusters and porous scaffold design.

However, heat dissipated in thin NOA63 substrate devices efficiently avoided exciton annihilation and sustained or improved device efficiency (Fig. 5a). This was consistent with the time-resolved EL (TREL) measurement, which reveals that the exciton lifetimes from thick glass and NOA63 devices were longer with a monotonic decay than those of thin NOA63 devices with double-exponential features (Fig. 5e and Supplementary Fig. 15). The TREL results showed overlapping overall, except that of glass-based device, and the detailed comparison could be confirmed (Supplementary Fig. 15b). On the other hand, although the thickness of the NOA63 containing NPs (NPs 4.5 wt%, 16.3 μm thickness) was twice that of the film without NPs (NOA63 9.2 μm), their identical TREL results (Supplementary Fig. 15c) indicate that the effect of thicker film in dissipating heat was minimized by the presence of NPs. It was reported that Aluminum Oxide (Al$_2$O$_3$) NPs dissipated heat efficiently from perovskite solar cells due to their highly, thermally conductive (20–30 W m$^{-1}$ K$^{-1}$) metal oxide NPs and porous scaffold design (the schematic illustration of Fig. 5e inset)[35]. Since the thermal conductivity of SiO$_2$ NPs (1.3 W m$^{-1}$ K$^{-1}$)[61] is much higher than NOA63 (0.2 W m$^{-1}$ K$^{-1}$), it can be inferred that SiO$_2$ NPs also facilitate heat dissipation from the device in a similar manner. In addition, the TREL of the device with thin elastomer (100 μm) appeared between those with 70 and 210 μm NOA63 based devices without elastomer, which means that the thin elastomer efficiently dissipated the heat from the inside of the device. Consequently, the total thickness of the substrate (polymer film and elastomer) was dominant in the heat dissipation mechanism. The heat-dissipation performance facilitated by thin elastomer and SiO$_2$ NPs were found to be effective as compared to other

techniques (Supplementary Table S2). The efficient heat dissipation process might disturb any free-carrier (excitons) interactions, such as TTA, because exciton recombination occurring in lower energy states (trapping sites) becomes dominant in an environment with lower thermal energy[32–34]. We believe that this observation is evidence for supporting the heat dissipation mechanism for thin NOA63 devices, where the faster heat exit, reduced the TTA process[32–34]. Therefore, a thinner elastomer is preferable to avoid heat accumulation and exciton annihilation can be further suppressed. That is why the phosphorescent OLEDs with thick glass substrate show the efficiency roll off at high exciton densities as compared to the NOA63 devices, which indicates that the lower substrate thickness is preferable to avoid it.

**Uniform emission without shifts in color coordinates.** Microcavity-based wavy/buckled OLEDs using a semi-transparent electrode present shifts in the wavelength or color coordinates between the unstretched (buckled) and stretched (planar) modes[27–29,39]. The microcavity effect was only observed for the semi-transparent Ag electrode (Fig. 6a), which potentially results in a chromic shift under various applied strains. The microcavity effect was negligible for the MAM electrode (Fig. 6b) because of the high transparency of the electrode. Further, when SiO$_2$ NPs were incorporated into the NOA63 substrate, the extraction of more photons was aided by the forward Mie scattering[54,55], thus increasing the efficiency (Fig. 6c). This mechanism for Ag and MAM was evaluated using the EL and International Commission on Illumination (CIE) 1931 color

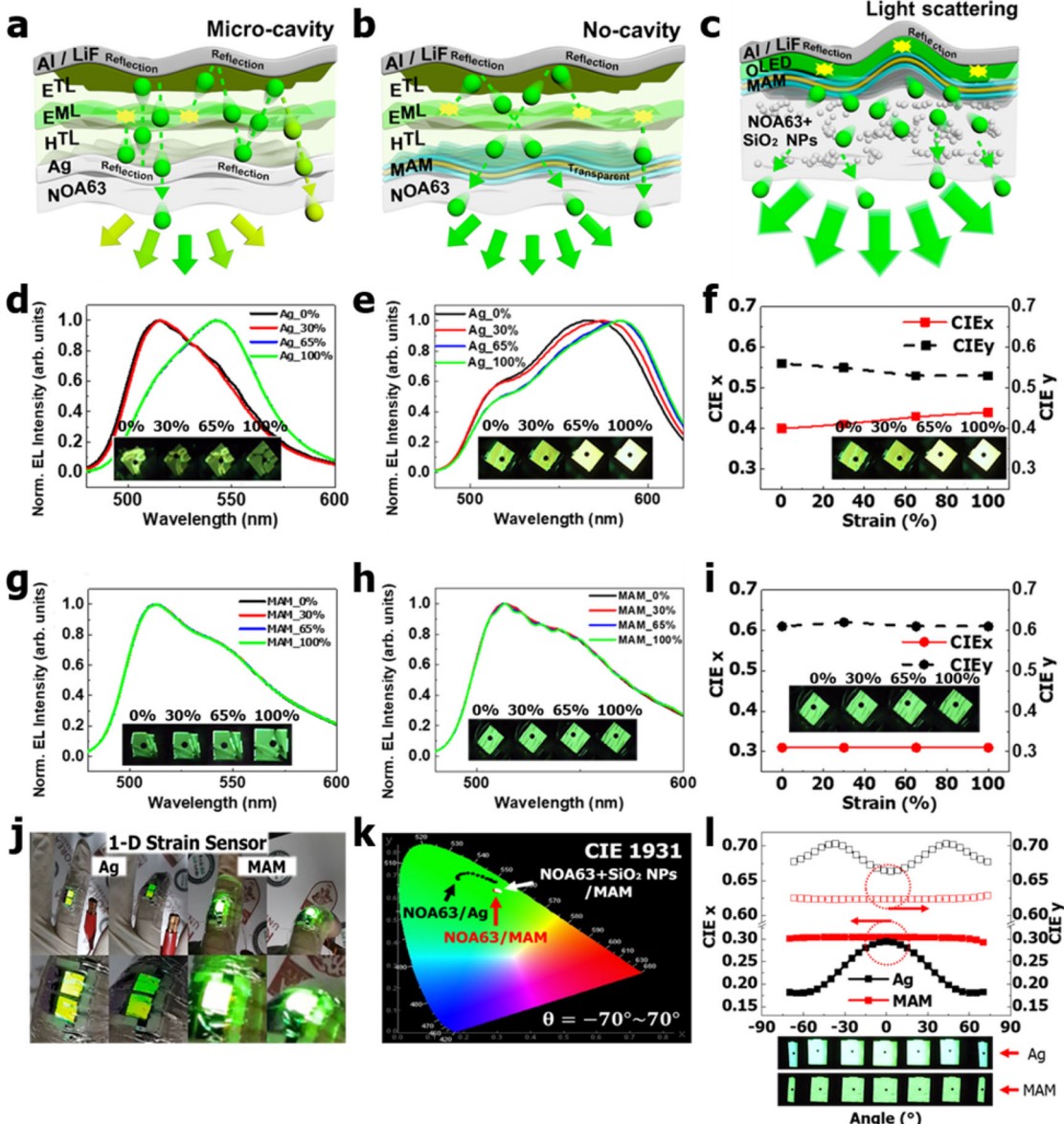

**Fig. 6 Mechanism of optical light scattering along with the EL and color coordinate analysis. a–c** Schematic illustration of the optical waveguide demonstrating the difference between light pathways for silver (Ag), MoO₃/Au/MoO₃ (MAM) and MAM with silicon dioxide nanoparticles (SiO₂ NPs) substrates. **d, e** Electroluminescence (EL) analysis of the devices with Ag electrodes under two-dimensional (2D) and one-dimensional (1D) stretching, respectively. Inset, the camera images of the relevant pixels from the devices. **f** The changes in the International Commission on Illumination (CIE)-1931 measurements for $x$ and $y$-coordinates using 1D stretching at various values (0–100%). **g, h** EL analysis of the devices with MAM electrodes under 1D and 2D stretching, respectively. Inset shows the camera images of the relevant pixels from the devices. **i** The CIE-1931 for $x$ and $y$-coordinates using 1D stretching at various strain percentages (0–100%). **j** Camera images for 1D strain sensor showing color shift for Ag-based devices, while no color shift for MAM-based devices. **k, l** CIE-1931 shifts for Ag and MAM electrodes at various measurement angles (−70°–70°) along with the comparison of CIE-1931 for Norland Optical Adhesive 63 (NOA63), with and without SiO₂ NPs/MAM and Ag devices.

coordinate changes of devices (Fig. 6d–i, Supplementary Fig. 16 and 17). The 2D stretching of the devices with the Ag electrode (Fig. 6d) showed EL shifts to longer wavelengths with increasing applied strain. The EL peaks were in two groups: one with 0 and 30% stretching at ~510 nm and the second group with 65 and 100% stretching, which was yellow-shifted towards ~550 nm. A more consistent trend by periodic buckling was observed for 1D stretching (Fig. 6e) at various percentages (0–100%) and the color change was observed by the camera images of the Ag device pixels (inset of Fig. 6e). There was also a yellow shift at 0% stretching (2D buckling) due to the change in the light path caused by random 2D and 1D periodic buckling[27–29,39]. Figure 6f shows the

$x$- and $y$-axis CIE 1931 changes for 1D stretching. With an increase in the applied strain, an increase in the x-CIE but a decrease in y-CIE indicates a yellow shift. In contrast, MAM electrode devices presented an initial emission at ~510 nm and CIE-1931 coordinates, which remained unchanged (Fig. 6g–i, inset of i and Supplementary Fig. 17) irrespective of the amount or the direction (1D, 2D) of the applied strain.

The shift in the color of the Ag electrode devices by the applied strain was encouraging for a wearable strain sensor[62] (Fig. 6j, Supplementary Movies 7 and 8). The Ag device attached to the straight position of the finger presented a yellow color, shifted to green upon bending, which was contrary to the green-to-yellow

shift in the EL spectra of the devices due to the observer's view angle. The relationship between the changes in the x-CIE, y-CIE, and the applied strain was formulated by equation (Supplementary Fig. 18). However, in MAM-based devices, the green emission by the device remained unchanged. This was also evaluated by the variable angle (−70°–70°) CIE 1931 emission analysis (Fig. 6k, l, Supplementary Figs. 16 and 17) for Ag and MAM devices. Thus, the MAM electrode was an optimum candidate for uniformly emitting GSOLEDs. Also, The full-wave optical simulations using the finite element (FE) method have been performed for the dynamics of the shifts in EL and the out-coupling efficiency enhancement (Methods and Supplementary Figs. 19 and 20) in various device structures, which follows the Fabry–Perot microcavity equation[63]. Further, the COMSOL analysis showed that better out-coupling efficiency resulted in better photon extraction with a pitch of 200 nm than that of 800 nm. The rougher surface of NOA63 by the bumpy surface modification and agglomeration of NPs by the UV treatment[35,44,45] with a 200 nm pitch could support the light extraction by scattering more efficiently than the smoother surface[64]. These show the proposed method in this work is suitable for the development of efficient GSOLEDs for stretchable/flexible and wearable, optoelectronic, and biomedical applications.

## Discussion

This research has demonstrated the dynamics of a bottom-emission GSOLED using thin Ag and MAM electrodes on a flexible/stretchable elastomeric substrate of NOA63/3 M elastomer, with and without SiO₂ NPs. The devices were conveniently fabricated using the modified kinetic transfer printing method. A significant improvement in the device performance with CE 82.4 cd A⁻¹, PE 57.5 lm W⁻¹, and EQE 22.3% was achieved using the MAM electrode with 470 nm SiO₂ NPs and a horizontal emitter. This was due to appropriate charge injection and recombination in the EML, because of better φ provided by the MAM electrode. The minimization of TTA using the heat dissipation mechanism, meant that there was no noticeable efficiency roll-off by the NOA63 devices (with and without NPs). In addition, we have confirmed that the SiO₂ NPs are beneficial for both optical out-coupling and the heat dissipation mechanism. Also, we have demonstrated the "Stretchable & Heat dissipating device" with a thin elastomer (100 μm). Further, the devices with Ag electrodes presented a chromic shift in the EL/CIE-1931 color coordinates, however, no chromic shift in the EL/CIE-1931 in the MAM devices was observed because of the higher transparent electrode and insignificant microcavity effect. The performances were cross-evaluated with optical and mechanical simulations, which further confirmed that the proposed method has developed efficient GSOLEDs for stretchable/flexible and wearable, optoelectronic, and biomedical applications.

## Methods

**Substrate cleaning and surface treatment**. The glass substrate was cleaned using sonication with acetone and isopropyl alcohol for 10 min each, followed by nitrogen drying. The substrate was dipped (~2 min) into a water-repellent detergent (Bullsone, South Korea) for surface modification to decrease the adhesion with the overlying layer for a convenient peel-off process.

**OLED device fabrication with MAM electrode**. A transparent and flexible (NOA63) layer was spin-coated on the water-repellent detergent treated glass at 5000 rpm for 20 s followed by UV curing at 187 nm wavelength for 5 min to attain a thickness of ~9.2 μm. For efficiency enhancement, 470 nm SiO₂ NPs (4.5 wt%) were mixed with NOA63 by stirring at 20 °C and 100 rpm for 2 h, followed by placing the solution in a desiccator for 3 h to remove bubbles. The SiO₂ incorporated NOA63 solution was spin coated using the same parameters as mentioned above. A MAM electrode with the configuration MoO₃(15 nm)/Au (14 nm)/MoO₃(5 nm) was deposited on the pristine, and SiO₂ incorporated NOA63 substrate using a thermal evaporator at a base pressure of 2.0 × 10⁻⁷ Torr.

N,N-bis-(1-naphthyl)-N,N′-diphenyl-1,1′-biphenyl-4,4′-diamine (NPB) as the HTL and tris (4-carbazoyl-9-ylphenyl) amine (TCTA) as an exciton blocking layer (EBL) were deposited with a thickness of 40 and 10 nm, respectively. The emission layer (EML) consisted of CBP with Ir(ppy)₃ (15 nm, 8% doping) and 2,2′,2″ -(1,3,5-benzinetriyl)-tris(1-phenyl-1-H-benzimidazole) (TPBi) with a thickness of 40 nm, was used as an electron transport layer (ETL). Lithium fluoride (LiF) (1 nm)/Al (100 nm) layers were deposited as a cathode using a shadow mask (25 mm²) with constant deposition rates of 0.5 Å/s and 5 Å/s, respectively. The fabricated devices were encapsulated with NOA63 (9.2 μm) or epoxy resin (UV RESIN XNR 5570B1, Nagase ChemteX Corporation). The reference OLEDs were fabricated with the same device structure for both MAM and Ag (20 nm) electrodes using thermal evaporation. For the passivation, SiN$_x$ was deposited using the radio frequency (RF) sputtering method 800 W (1 W cm⁻²). Si₃N₄ target for 30 min at a base pressure of 5 μTorr and working pressure of 1 mTorr. The distance between the target and the NOA63 substrate was 160 mm.

**Buckling mechanism with a biaxially pre-strained elastomeric substrate**. A 3 M VHB elastomer (3 M, USA) and a thin elastomer (DOW Corning HTV processed by Apple Silicone, Korea) were utilized as the receiving substrate (25 cm²). The edges of the 3 M elastomer were clamped in the jaws of a custom-made mechanical stretching jig, and the 3 M elastomer was biaxially strained up to 100 cm² (400% of the original area). The fabricated OLED devices along with the NOA63 film were conveniently peeled from the glass substrate due to the hydrophobic nature of the surface energy, provided by the detergent treatment of the glass surface (mentioned before) and were transferred to the pre-strained 3 M elastomer. The removal of strain caused shrinkage of the 3 M elastomer, which led to the formation of buckles in the device, as shown in Fig. 1.

**IR-camera measurement**. An IR-camera (E5-XT, FLIR systems, Inc., USA) was used to measure the temperature on the substrate due to the heat generated by the OLED. The target temperature range is −20 °C to 400 °C and the resolution of the IR-camera is 160 × 120 (19,200 pixels). The emissivity of the glass and epoxy resin (NOA63) was 0.95. The noise equivalent temperature difference (NETD) of the IR-camera was <0.1 °C.

**Pulse I–V measurement**. The pulse I–V measurement was completed using a Keithley 4200 semiconductor characterization system (SCS) and a Keithley 4225 pulse measurement unit (PMU) with a rectangular pulse width $W_{\text{pulse}} = 10$ μs and 50% duty cycle.

**OLED device performance**. The J-V-L and EL analysis for planar, stretched (0–100%), twisted, flexible, and water immersed OLEDs was conducted using Keithley 2400 voltmeter and Minolta CS-2000 spectroradiometer. The variable angle luminance from the devices was analyzed for Lambertian emission using a 1-axis control box with a rotating jig (Jaewon, South Korea).

**Optical properties (transmittance and absorption)**. The transmittance/absorbance of the substrates was measured using a UV/vis/near-IR spectrophotometer (JASCO V-570).

**Conductivity analysis**. The MAM electrode layers were deposited on a glass substrate for conductivity analysis using 4-probe source measurement units (Keithly 236 source, 700 switch system, 6485 picoammeter by Jeong Yeon Systems, Korea).

**Surface morphology and roughness**. The surface morphology of the deposited layers was analyzed using a UHR FE-SEM, S-5500, Hitachi, at the Korea Basic Science Institute (KBSI) Jeonju Center with an accelerating voltage of 7 kV.

**UPS analysis**. UPS spectra (VG, ESCALab MKII, UK) were obtained with the samples biased at −10.0 V to clear the secondary cutoff. An energy resolution of 0.02 eV from the slope of the Fermi edge of the MAM electrode was utilized.

**Optical microscopy**. OM analysis of the devices was conducted using an OM (KSM-BA3T, Samwon Scientific Ind. Co. Ltd. Korea) to analyze the surface morphologies of the samples.

**Confocal microscopy**. The CM images were taken using a confocal microscopy tool (vk-x1000 laser, vk-x1050 stage, Keyence, Osaka, Japan). The characterization tool is equipped with a class 2 laser (661 nm) with high-definition differential interference color images (16-bit).

**TREL measurement**. A function generator (9205C, Protek) was used to inject periodic TTL square waves (155.6 kHz repetition rate and 50% duty cycle) into an EL device, and subsequent EL was collected using an optical fiber (400 μm core). The EL signal was sent to a monochromator (SP-2150, Princeton Instruments,

Inc.) and detected with a single-photon avalanche photodiode (ID-100-50, IDQ Inc.). A commercial time-correlated single-photon counting (TCSPC) board (SPC-130-EMN, Becker & Hickl Inc.) was employed to record transient EL profiles in a 5 μs window. TREL spectra were recorded by automatically scanning the mono-chromator position. Truncated TREL profiles, that are limited above the maximum intensity, were fitted with multi-exponential functions.

**Lifetime measurement**. The lifetime measurement of the devices was performed at a constant temperature and a humidity of 25 °C and 25%, respectively using a photodiode based lifetime measurement system (Polaronix M6000, McScience) until the luminance of the devices dropped to 50% of the initial luminance for LT50 while the devices were operated at a constant current density. Lifetime under various strains was conducted with a custom-made mechanical stretching jig in a dark box using Keithley 2400 voltmeter and Minolta CS-2000 spectroradiometer. Also, for the water immersion test of the devices with various encapsulations, the lifetime of the devices was measured with the same equipment keeping the devices in the water until the luminance became zero (LT0).

**Optical simulations**. We performed full-wave optical simulations using the FE method (COMSOL Multiphysics, RF module) to understand the dynamics of the shifts in the emission wavelength (Supplementary Fig. 19a, b, d, e) and the out-coupling efficiency (Supplementary Fig. 19f) in various device structures. For the analysis of the emission spectrum in Supplementary Fig. 20a- f, the GSOLED with the 20 nm thick Ag electrode and the 34 nm thick MAM electrode (5 nm thick $MoO_3$, 14 nm thick Au, and 15 nm thick $MoO_3$ layers) were used. For the analysis of the out-coupling from NOA63 substrate with and without NPs (Supplementary Fig. 20g–i), we assumed that periodic bump structures on NOA63 substrate with a height of 50 nm and a width of 148 nm were formed (Supplementary Fig. 19i). For all simulations, we assumed incoherent point dipole sources in the emissions.

**Mechanical simulations**. A commercial software (ABAQUS v 6.14-1) was used to simulate the 2D stretching of the devices. To reflect the actual wrinkle shapes in the device, linear buckling analysis was completed. Based on the results of the buckling analysis, the stress distribution change in the wrinkled OLED was analyzed when the biaxial strain was applied to the elastomeric substrate. The device was dis-cretized into quadratic brick elements with reduced integration (C3D20RH). Multilayers in the device were simplified to three materials (OLED, NOA63, 3 M elastomer), and the elastic modulus of each material in the film device was assumed based on the literature[8,9,56].

## Data availability

The data that support the plots within this paper and other findings of this study are available from the corresponding author upon reasonable request.

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

## Acknowledgements

This research was supported by Basic Science Research Program through the National Research Foundation of Korea (NRF) funded by the Ministry of Education (NRF-2014R1A6A1030732, 2020R1A2B5B01001580). This work was supported by "Human Resources Program in Energy Technology" of the Korea Institute of Energy Technology Evaluation and Planning (KETEP), granted financial resources from the Ministry of Trade, Industry & Energy, the Republic of Korea (No. 20204030200070). This work was also supported by the Future Growth Engine Program (10079974, Development of core technologies on materials, devices, and processes for TFT backplane and light-emitting front plane with enhanced stretchability above 20%, with application to stretchable display) funded by the Ministry of Trade, Industry & Energy (MOTIE, Korea). We also acknowledge support from the fundamental research program (PNK7400) of the Korea Institute of Materials Science (KIMS). This work was supported by the Technology Innovation Program (20010804, Development of solution type polarizing materials and thin-film circular polarizer for flexible OLED applications thickness below 30 μm transmittance above 41% and polarization efficiency above 98%) funded by the Ministry of Trade, Industry & Energy (MOTIE, Korea). And, this research was supported by the BK21 FOUR (Fostering Outstanding Universities for Research) funded by the Ministry of Education (MOE, Korea) and the National Research Foundation of Korea (NRF).

## Author contributions

The manuscript was prepared through contributions from all the authors. S.Y.R., C.S.K., and H.H., the corresponding authors, designed and conceived this work. D.K.C., D.H.K., and C.M.L. fabricated all GSOLEDs and completed all measurements. H.H. and S.S. wrote the manuscript and organized all analyses with M.K.S., D.H.K., C.M.L., H.J.C., G. W.J., T.W.K., J.S.Y., and D.H.C. provided ideas on the fabrication and tests for GSOLEDs with stretchable jig and SiO₂ NPs incorporated into the NOA63 film. S.Y. and J.S. suggested an analysis of the optical scatterance calculation and temperature analysis by an IR-camera. T.S.K. and B.S.M. conducted and analyzed the mechanical simulation of the GSOLED according to stretching 3 M elastomer. C.H.K. and H.J.L. built the TCSPC instrument integrated with the monochromator, and performed a TREL experiment on GSOLEDs. J.W.L. and D.K. tested the GSOLEDs by pulse and DC operation. T.S.B. and S.M.Y. characterized the FE-SEM images of NOA63 with and without NPs. Y.C.K. and J. P. coordinated the UPS analysis on the MAM and Ag electrodes. K.H.K. and M.S. conducted an optical simulation of the EL shift from the Ag and MAM electrodes and optical enhancement by NP pitch. All authors read and have given their approval of the final version of the manuscript.

## Competing interests

The authors declare no competing interests.
