## [Peer Review File · Nature Communications]

REVIEWER COMMENTS

Reviewer #1 (Remarks to the Author):

The authors achieved highly efficient/stretchable OLED by using MOM as a transparent electrode and ultra-violet-curable polymer, NOA63/3M elastomer, as substrate. Especially, by doping SiO₂ NPs into NOA63, which has good heat dissipation characteristics, and by reducing the reflectance of the substrate, the device efficiency could be improved. These devices exhibited a high current efficiency of 82.5 cd/A and EQE of 22.6% with minimum efficiency roll-off, which didn't show any change of CIE coordinate after stretching. Electrical and optical characterization are well organized and are in good agreement with experimental results.

Therefore, the reviewer suggests the publication of this paper to 'Nature communications' after major revision.

1. Please include the device lifetime results according to the encapsulation method to check the encapsulation characteristics. Also, please characterize the device lifetime after water immersion (in figure 2).
2. Please add TrEL data of sample with "NOA63+NPs" to check the exciton recombination mechanism after doping.
3. The authors demonstrate that the device (heat dissipated device, Figure 3b) showed no critical efficiency roll-off issue. However, the device itself is not stretchable at all without the 3M tape, which is in conflict with the stretchable application.
4. The author uses the heat dissipated device for efficiency record and use the non-heat dissipated device for stretching test. It is too tricky, so please correct it in the revised manuscript.
5. Papers on two-dimensional stretchable LED have already been published. (ACS Appl. Mater. Interfaces 2016, 8, 31166–31171, Organic Electronics 2017, 48, 314). Please discuss and compare with the above-mentioned paper and notify the novelty of this research.

Reviewer #2 (Remarks to the Author):

Comments to the Authors

The authors demonstrated a highly efficient, heat-dissipating, geometrically twistable, stretchable organic light-emitting diode (GTS OLED) based on thin oxide/metal/oxide electrodes, pre-strained elastomer, and thin ultra-violet curable polymer encapsulation layer with silicon dioxide nanoparticles.

Here, to have stretchability, the authors used pre-strained 3M elastomer to make random wavy buckles. A thin oxide-metal-oxide electrode and thin ultra-violet curable polymer can be under strain based on random wavy buckles even though they have high young modulus. Using the transparent oxide/metal/oxide electrodes shows uniform emission without chromic shift under stretching, and using the silicon dioxide nanoparticles in the encapsulation layer achieves heat-dissipating properties.

This work builds on previously well-developed technologies around geometrically stretchable electronics, which means general ideas used for geometrically stretchable organic light-emitting diodes are not completely new. Also, the ultra-violet curable polymer, especially for a Norland Optical Adhesive 63, has been used for encapsulation in stretchable OLEDs (ref. 18-20). The silicon dioxide nanoparticles are used to added to the NOA63 film to help light scattering to enhance the outcoupling efficiency (ref. 32). However, the authors systematically integrated the mentioned technologies and fabricated GTS OLEDs with enhanced electrical performance, and heat dissipation, stretchability, and uniform emission without chromic shift. The authors also provide the mechanism and dynamics of fabricated GTS OLEDs based on oxide/metal/oxide electrodes.

I think the application level and stable electrical performance as a contribution to the field of stretchable electronics may draw some interest of readers of Nature Communications, but I do not think the current version reaches the appropriate level for publication -- major revisions in re-organizing the content and adding more detailed discussion according to the major comments and questions below are required.

Major comments and questions

#1. The authors should address heat dissipation technologies for the organic light-emitting diodes and flexible/stretchable electronics in the introduction. In addition, the manuscript should include

additional details and reviews on oxide/metal/oxide electrodes.

#2. The authors mentioned that geometrically stretchable OLEDs (GSOLEDs) were previously introduced, and this research realized geometrically twistable, stretchable OLEDs (GTS OLEDs). What is the difference between previously GSOLEDs and these GTS OLEDs?

#3. There are two full names of GTS OLED, one is geometrically twistable, stretchable organic light-emitting diodes (Line 38), and the other one is geometrically twistable, stable, stretchable organic light-emitting diodes (Line 90). It would be nice to unify as one.

#4. The authors show the images of the device performance during twisting and stretching tests and water immersion in Figure 2. The camera images confirm that the devices work well under various operation modes, but there is no numerical data for how much performance degradation occurs when twisting or stretching and how well restoration is performed. The data at various applied 2D strains (0-100%) are in Table 1 and Figure 3, but there is no data for twisting mode. Furthermore, the label's uniformity attached to Figure 2a-c, (a: Camera Images, Stretching Images, Operation Test) should be improved.

#5. In Figure 2, there is no notification about the samples that incorporate NOA63 film with NPs or without NPs both in text and figure caption.

#6. Figure 3 does not clearly show the device performance with and without SiO₂ NPs and the heat sink mechanism. I notice that the Scheme in Figure 3a is the full device structure of GTS OLEDs. The authors show no critical efficiency roll-off issue through Figure 3c-d, even when the 3M elastomer does not allow efficient heat dissipation. Figures 3b, e-f are designed to show the effects of thin NOA63 film with NPs by comparing glass-based, NOA63 film only, and NOA63 film with NPs based on non-GTS devices without thick 3M elastomer.

First, the meaning of the labels 'heat accumulation' and 'heat sink or dissipation' in Figures 3a, b are not clear.

Second, in figure 3c-d, the readers will be unable to appreciate the heat sink effects of using NOA63 film with or without NPs rather than other encapsulation layers.

Third, Figure 3b, e-f and 4 compares the heat dissipating properties of thin NOA63 film and thick glass substrates. How about the heat dissipating properties when the devices use thin glass substrates or other thin, flexible, and stretchable substrates?

Fourth, the authors said that Ir(ppy)₃ was used as a dopant, but to achieve a high device efficiency Ir(ppy)₂ tmd is also used and analyzed. Why is there a difference between using Ir(ppy)₂ tmd for Figure 3a and Ir(ppy)₃ for Figure 3b?

#7. At a 50% stretching strain of 3M tape, the OLED devices' overall strain level was approximately 3%. How much the overall strain level of OLED devices under a 100% stretching strain of 3M tape, and is it below the critical level of materials in OLED devices?

#8. The full information of reference 14 is missing.

Minor comment on typographic errors.

#1. Check Ref. 41, 50.

Reviewer #3 (Remarks to the Author):

The authors reported geometrically stretchable OLED by the NOA63 substrates and MAM electrodes. The device showed 100% stretchability by being laminated onto 1mm thick elastomer, and stability in mechanical strain and water immersion. Furthermore, the authors investigated the effect of substrate thickness on the efficiency roll-off, and stability of EL intensity over strain with Ag and MAM electrodes. However, the paper shows poor novelty as research. For example, stretchable OLED using buckling is already demonstrated. The advantage of MAM over other transparent electrodes (e.g. PEDOT:PSS, ITO, AgNWs) are not shown. Although the reviewer evaluate the investigation on the effect of substrate thickness and microcavity effect, the novelty

is not enough for Nature Communications. Besides, the authors do not cite papers properly. Therefore, the reviewer recommends the rejection of this manuscript from Nature Communications. The followings are the detailed comments.

1. As authors cite, geometrically stretchable OLED has been demonstrated in many other literatures. Besides, the substrate thickness of 9.2 μm is very thick compared with the previous reports where they are using 2-3 μm thick plastic as a substrate. This thick substrate thickness caused very large buckling periodicity ($\sim 500 \mu\text{m}$), which makes it difficult in the application due to the bad appearance. NOA with SiO_2 is 16.3 μm , which makes the appearance as "stretchable OLED" even worse.

2. Geometrically stretchable OLEDs using transparent electrodes (PEDOT:PSS, ITO, AgNWs) have been already demonstrated. The authors should clearly show the advantage of MAM over them.

3. WVTR of NOA with 3M tape is reported. However, considering the thickness, it is not impressive.

4. The followings are the examples of "not properly cited" references. The reviewer strongly recommend to revise before submitting elsewhere.

Page 1 "However, these methods have drawbacks including poor uniformity, process complexity, low conductivity, instability, low transmittance, and work function (ϕ) mismatches for carrier injection" has no reference or evidence to prove.

Reference 5 has nothing to do with the hygroscopicity of polyurethane.

Reference 14 is not cited properly.

Ref 26 is not intrinsically stretchable OLED.

REVIEWER COMMENTS

Reviewer #1

Recommendation: Major Revision

The authors achieved highly efficient/stretchable OLED by using MOM as a transparent electrode and ultra-violet-curable polymer, NOA63/3M elastomer, as substrate. Especially, by doping SiO₂ NPs into NOA63, which has good heat dissipation characteristics, and by reducing the reflectance of the substrate, the device efficiency could be improved. These devices exhibited a high current efficiency of 82.5 cd/A and EQE of 22.6% with minimum efficiency roll-off, which didn't show any change of CIE coordinate after stretching. Electrical and optical characterization are well organized and are in good agreement with experimental results. Therefore, the reviewer suggests the publication of this paper to 'Nature communications' after major revision.

Comment 1: Please include the device lifetime results according to the encapsulation method to check the encapsulation characteristics. Also, please characterize the device lifetime after water immersion (in figure 2).

Authors Reply: Thank you for your kind comment. We have tried with improved encapsulation methods to achieve longer device lifetime. We have conducted the device lifetime measurement (LT50) and water immersion test (LT0). In addition, the water vapor transmission ratio (WVTR) by the calcium (Ca) test for various encapsulations and substrates was analyzed as shown below.

- **Device A:** NOA63 / 3M tape / NOA63
- **Device B:** NOA63 / 3M tape + NOA63 side passivation / NOA63
- **Device C:** NOA63 / SiN_x / 3M tape + NOA63 side passivation / SiN_x / NOA63 / SiN_x
- **Device D:** NOA63 / SiN_x / 3M tape + epoxy side passivation / SiN_x / NOA63 / SiN_x

The LT50 measurement was performed at a constant temperature and humidity (25 °C and 25%), respectively until the luminance of the devices dropped to 50% of the initial luminance (1000 cd/m²). The LT50 of device **A** showed 75 h, which increased to 90 h for those of devices with NOA63 side passivation (**B** and **C**). For the epoxy (UV RESIN XNR 5570B1, Nagase ChemteX Corporation) side passivation (**D**), the LT50 increased further to 110 h.

Moreover, the LT0, similar with LT50 but measured until 0% luminance, suggests that side passivation is critical to the device stability and the device lifetime in water immersion test. When a SiN_x thin film adapted as passivation, it showed almost twice compared with NOA63 passivation (**B**). The device remained intact when epoxy side passivation (**D**) was applied. Thus, in the harsh condition like water immersion, the most important factor is side encapsulation.

The values of WVTR obtained for the samples are summarized in the table of Figure S3. From the WVTR result, we speculate that the 3M tape alone as a sealant was not the proper material to use for the purpose, as it could have allowed moisture to penetrate into the substrate. Comparing WVTR for **C** and **D**, epoxy was found to be better than NOA63 in blocking the side from the moisture. However, the WVTR value of the device **C** ($3.6 \times 10^{-2} \text{ g}\cdot\text{m}^{-2} \text{ day}^{-1}$) is comparable to that of 4-6 layers of graphene-encapsulation ($1.78 \times 10^{-2} \text{ g}\cdot\text{m}^{-2} \text{ day}^{-1}$).

Thus, it may be concluded that the side encapsulation is as important as encapsulation against penetration through top and bottom layers. Even though the side encapsulation with epoxy was quite effective, there is a trade-off between stretchability and encapsulation. It needs further study to improve the encapsulation with stretchability for GSOLED.

(Page 38, Figure 4 in the revised manuscript)

(Page 11, line 15 ~ Page 12, line 9 in the revised manuscript)

“As suggested in the Figure 2, the device lifetime (LT) under various encapsulation schemes (Device A : NOA63/only 3M tape side passivation/NOA63; Device B : NOA63/3M tape and NOA63 side passivation/NOA63; Device C : NOA63/SiN_x/3M tape and NOA63 side passivation/SiN_x/NOA63/SiN_x; Device D : NOA63/SiN_x/3M tape and epoxy side passivation/SiN_x/NOA63/SiN_x) and water immersion condition have improved and were comparable, respectively (shown in Fig. 4c, d). All of data trend was found to be coherent with the conditions. It may be concluded that the “side passivation” from a combination of 3M tape and NOA63 is quite critical as well as protective against “film penetration” through NOA63 films in a vertical direction. Device A encapsulation withstood for about 7 min during the water immersion, while Device B and C with NOA63 side passivation survived for over 1~3 hours. The shorter LT0 of the device A encapsulation as compared to its LT50 (Fig. 4c, d) is due to side penetration of water, which was confirmed by WVTR measurement (Supplementary Figure S3). Even though the side encapsulation with epoxy (Device D) was quite effective (Fig. 4d), there is a trade-off between stretchability and encapsulation. It needs further study to improve the stretchable encapsulation for GSOLEDs. Additionally, the LT50 at the different strain conditions (0%, 30%, 65%, 100%) was also analyzed with 3M elastomer and the tendency remained the same despite of the strain. It means that geometrical stretching does not affect the device operation due to the mechanical consideration.”

(Page 6, Supplementary Figure S3 in the revised supplementary information)

$$\begin{aligned}
 \text{WVTR} &= \rho(\text{CaO}) \times \frac{m(\text{H}_2\text{O})}{m(\text{CaO})} \times \frac{D(\text{CaO})}{t} \quad [\text{g/m}^2\text{day}] \\
 &= 3.35 \times 10^6 \text{g/m}^3 \times \frac{18.01528 \text{g/mol}}{56.077 \text{g/mol}} \times D(\text{CaO})/t \quad [\text{g/m}^2\text{day}] \\
 &= 1.07622 \times 10^6 \text{g/m}^3 \times D(\text{CaO})/t \quad [\text{g/m}^2\text{day}] \\
 &\Rightarrow 1.07622 \times 10^6 \text{g/m}^3 \times \left(\frac{T(\text{Final}) - T(\text{Initial})}{100 - T(\text{Initial})} \times 500 \times 10^{-10} \text{m} \right) / \left(\text{Time}(\text{min}) \times \frac{1\text{h}}{60\text{min}} \times \frac{1\text{day}}{24\text{h}} \right)
 \end{aligned}$$

* ρ : Density, m : Molar concentration, D : Thickness, t : Time, T : Transmittance

Comment 2: Please add TrEL data of sample with “NOA63+NPs” to check the exciton recombination mechanism after doping.

Authors Reply: Thank you for your valuable comment. We have updated the manuscript comparing the TREL data of the devices with and without NPs to check the exciton recombination mechanism, again. We have concluded from the TREL result that the incorporation of SiO₂ NPs with NOA63 improves the heat dissipation of the devices. It has been already reported that metal oxide NPs dissipated heat efficiently in optoelectronic devices due to their high thermal conductivity and porous scaffold design. [Energy & Environmental Sci., 13, 5059-5067 (2020)] Even though NOA63+NPs (16.3 μm) substrate was thicker than NOA63 (9.2 μm), almost similar TREL results suggested that NPs facilitated the heat dissipation mechanism due to the thermally conductive NPs (1.3 W m⁻¹ K⁻¹) compared to the NOA63 (0.2 W m⁻¹ K⁻¹) and porous scaffold design.

(Page 40, Figure 6e in the revised manuscript and Page 33, Supplementary Figure S18b in the revised supplementary information)

(Page 13, line 11 ~ Page 14, line 17 in the revised manuscript)

“The generated and accumulated heat in thick substrate devices provided thermally activated energy to triplet excitons, which boosted longer exciton lifetimes, TTA, and TPA, finally degrading the device efficiency at high exciton density.^{29, 30, 31, 32, 33} However, heat dissipated in thin NOA63 substrate devices efficiently avoided exciton annihilation and sustained or improved device efficiency (Fig. 6a). This was consistent with the time-resolved EL (TREL) measurement, which reveals that the exciton lifetimes from thick glass and NOA63 devices were longer with a monotonic decay than those of thin NOA63 devices with double-exponential features (Fig. 6e and Supplementary Figure S18). The TREL results showed overlapping overall, except that of glass based device and the detailed comparison could be confirmed (Supplementary Figure S18b). On the other hands, although the thickness of the NOA63 containing NPs (NPs 4.5 wt%, 16.3 μm thickness) was twice that of the film without NPs (NOA63 9.2 μm), their identical TREL results (Fig. S18c) indicate that the effect of thicker film in dissipating heat was minimized by the presence of NPs. It was reported that Aluminum Oxide (Al_2O_3) NPs dissipated heat efficiently from perovskite solar cells due to their highly, thermally conductive ($20 \sim 30 \text{ W m}^{-1} \text{ K}^{-1}$) metal oxide NPs and porous scaffold design (the schematic illustration of Fig. 6e inset).³⁴ Since the thermal conductivity of SiO_2 NPs ($1.3 \text{ W m}^{-1} \text{ K}^{-1}$)⁵⁵ is much higher than NOA63 ($0.2 \text{ W m}^{-1} \text{ K}^{-1}$), it can be inferred that SiO_2 NPs also facilitate heat dissipation from the device in a similar manner. In addition, the TREL of the device with thin elastomer (100 μm) appeared between those with 70 μm and 210 μm NOA63 based devices without elastomer, which means that the thin elastomer efficiently dissipated the heat from the inside of the device. Consequently, the total thickness of the substrate (polymer film and elastomer) was dominant in the heat dissipation mechanism. The efficient heat dissipation process might disturb any free-carrier (excitons) interactions, such as TTA, because exciton recombination occurring in lower energy states (trapping sites) becomes dominant in an environment with lower thermal energy.^{31,}

^{32, 33} We believe that this observation is evidence for supporting the heat dissipation mechanism for thin NOA63 devices, where the faster heat exit, reduced the TTA process.^{31, 32, 33} Therefore, a thinner elastomer is preferable to avoid the heat accumulation and exciton annihilation can be further suppressed. That is why the phosphorescent OLEDs with thick glass substrate show the efficiency roll off at high exciton densities as compared to the NOA63 devices, which indicates that the lower substrate thickness is preferable to avoid it.”

Comment 3: The authors demonstrate that the device (heat dissipated device, Figure 3b) showed no critical efficiency roll-off issue. However, the device itself is not stretchable at all without the 3M tape, which is in conflict with the stretchable application.

Authors Reply: We appreciate your kind suggestion that the device without the 3M elastomer is not stretchable. Following your comments, we have tried to fabricate the GSOLEDs with thin elastomer (100 μm , DOW Corning HTV processed by Apple Silicon), expecting GS & heat dissipation at the same time. Initially, we have compared two kinds of device platforms. The first one was the “Stretchable platform” on thick 3M elastomer without heat dissipation function and the second one was the “Heat dissipation platform” only on NOA63 substrate without elastomer and without stretchable function.

During this revision, we have tried to combine previous two platforms, which means “Stretchable & Heat dissipation platform” on thin elastomer (100 μm) with stretchability and heat dissipation functions. We could achieve both properties in a single device but with some trade-offs as the third platform showed limited stretchability (30% strain) due to the too thin thickness of elastomer. Also, the heat dissipation property of the device was not as good as NOA63 because it has still 100 μm thickness. We have also tried to fabricate stretchable and heat dissipating devices with thinner elastomer (less than 100 μm); but, it was hard to be fabricated due to the limitations of the current techniques. However, we hope that in our future work we shall attempt a method to develop very thin elastomer (<100 μm), in which case the stretchability as well as the heat dissipation from the devices using our proposed fabrication will be much more enhanced.

(Page 11, line 1 ~ 14 in the revised manuscript)

“As we have discussed above, the stretchable devices could not show heat dissipation and heat dissipating devices could not be stretchable. To combine both qualities, additional devices have been fabricated using a thin elastomer (100 μm) at the different strain conditions (0, 10, 20 and 30%) as shown in Fig. 4. The heat dissipation function of the devices based on the thin elastomer were much efficient than that based on thick 3M elastomer, but less efficient than the NOA63 substrate only. Also, it was not as good as that of the device made only on thin NOA63 (9.2 μm) layers at low exciton density (below 10,000 nit) due to the thickness of the elastomer, but they performed similarly over 10,000 nits. The increment of CE with thin elastomer was a bit lower than that of only thin NOA63 substrate due to thin elastomer thickness (100 μm), which means that small quantity of heat still did not dissipate from thin elastomer. Moreover, the stretchability of thin elastomer ($\sim 30\%$) is much lower than that of 3M elastomer ($\sim 100\%$) because the thin elastomer could not sustain more than 30% pre-strain. We have also tried to fabricate stretchable and heat dissipating devices with even thinner elastomer (less than 100 μm); however, it was hard to be fabricated due to the handling difficulties of being torn over 30% strain.”

(Page 38, Figure 4a, b in the revised manuscript)

Comment 4: The author uses the heat dissipated device for efficiency record and use the non-heat dissipated device for stretching test. It is too tricky, so please correct it in the revised manuscript.

Authors Reply: We appreciate your concern regarding the differences of the device configurations in measuring efficiency and stretching test. In response to your comment, we fabricated new devices with a thin elastomer (100 μm), as we mentioned earlier (comment # 3), combining previous two platforms into one with stretchability ($\sim 30\%$ strain) and heat dissipation functions on high exciton density (over $\sim 10,000$ nit). Hence, we have corrected the manuscript accordingly.

(Page 11, line 1 ~ 14 in the revised manuscript)

“As we have discussed above, the stretchable devices could not show heat dissipation and heat dissipating devices could not be stretchable. To combine both qualities, additional devices have been fabricated using a thin elastomer (100 μm) at the different strain conditions (0, 10, 20 and 30%) as shown in Fig. 4. The heat dissipation function of the devices based on the thin elastomer were much efficient than that based on thick 3M elastomer, but less efficient than the NOA63 substrate only. Also, it was not as good as that of the device made only on thin NOA63 (9.2 μm) layers at low exciton density (below 10,000 nit) due to the thickness of the elastomer, but they performed similarly over 10,000 nits. The increment of CE with thin elastomer was a bit lower than that of only thin NOA63 substrate due to thin elastomer thickness (100 μm), which means that small quantity of heat still did not dissipate from thin elastomer. Moreover, the stretchability of thin elastomer ($\sim 30\%$) is much lower than that of 3M elastomer ($\sim 100\%$) because the thin elastomer could not sustain more than 30% pre-strain. We have also tried to fabricate stretchable and heat dissipating devices with even thinner elastomer (less than 100 μm); however, it was hard to be fabricated due to the handling difficulties of being torn over 30% strain.”

(Page 38, Figure 4a, b in the revised manuscript)

Comment 5: Papers on two-dimensional stretchable LED have already been published. (ACS Appl. Mater. Interfaces 2016, 8, 31166–31171, Organic Electronics 2017, 48, 314). Please discuss and compare with the above-mentioned paper and notify the novelty of this research.

Authors Reply: We are thanking you for kind comments. The table (Supplementary Table S1) below summarizes the novelty of our work as compared to the previous works presented in “ACS Appl. Mater. Inter. 2016, 8, 31166–31171” and “Organic Electronics 2017, 48, 314”.

The main configurational difference is in the anode, where ACS Appl. Mater. Inter. paper and Org. Electron. paper used Ag and Au, respectively but the present work used MAM as the anode for the first time in a GSOLED. Both works could not show invariance of the color coordinate and viewing angle after stretching. ACS Appl. Mater. Inter. work reported efficiency at 50% strain and Org. Electron. work reported at 3% strain; our present work suggests no efficiency roll-off and no shift in color-coordinate until 100% strain for 3M elastomer and 30% strain for thin elastomer. Moreover, the present work shows the current efficiency of 99.4 cd/A at 20,000 cd/m² for an encapsulated GSOLED with thin elastomer including heat dissipation function.

In summary, we would like to emphasize our strong points.

1. Demonstration of encapsulated GSOLEDs under strain condition and water immersion.
2. Application of SiO₂ NPs for optical out-coupling and heat dissipation, concurrently.
3. Realization of stretchability and heat dissipation properties with thin elastomer (100 μm), simultaneously.

Moreover, we have conducted two simulations: one is the mechanical simulation and the other one is the optical simulation. We found that thin NOA63 as a substrate dissipated heat efficiently and NOA63 with NPs was helpful for optical out-coupling with heat dissipation. Also, we have tried to fabricate devices with various encapsulation conditions and found that side encapsulation was tremendously critical for device stability. However, there is a trade-off relation between stretchability and encapsulation. So, it needs further study to improve the encapsulation with stretchability for GSOLED. In the light of the above points, we hope that the presented devices in this work demonstrate a step further towards the advancement of the next-generation GSOLEDs.

(Page 6, line 16 ~ 19 in the revised manuscript)

“The MAM-based twistable GSOLEDs with encapsulation showed high external quantum efficiency (EQE) and color stability at various strains and angles without efficiency roll-off until 100% strain that made the device unique compared to the other GSOLEDs (Supplementary Table S1).^{27, 38,}”

(Page 18, line 10 ~ 13 in the revised manuscript)

“A significant improvement in the device performance with CE 82.4 cd A⁻¹, PE 57.5 lm W⁻¹, and EQE 22.3% was achieved using the MAM electrode with 470 nm SiO₂ NPs and a horizontal emitter. This was due to an appropriate charge injection and recombination in the EML, because of better φ provided by the MAM electrode.”

(Page 4, line 2 ~ 6 in the revised supplementary information)

“Interestingly, when an outer molybdenum trioxide (MoO₃) layer (15 nm) was added before the deposition of Au (i.e. MA), the φ was increased to ~5.23 eV due to Fermi-Level pinning and the higher φ of the MoO₃ layer (~6.6 eV^{1, 2}). The φ of the electrode was further enhanced to ~5.25 eV by the deposition of an inner MoO₃ (5 nm) in addition to the MA structure (i.e. MAM) and is more beneficial for device performance, hence, improving the electrical and optical properties.^{3, 4, 5,}”

(Page 43, line 2 ~ 15 in the revised supplementary information)

“**a**, A schematic image of the buckled GSOLED device. **b**, Governing equation of linear buckling analysis in FE simulation. Buckling, such as wrinkles in thin films are unstable phenomena. This means that they are not implemented in FE simulation using a static solver because there is no load or moment component that will cause bending in the lateral direction. Therefore, we used

linear buckling analysis which can simulate wrinkled structures in the device.³⁸ First, the eigenvalue problem was solved in FE simulation with consideration of structure and mechanical properties of the device. The buckled shape of the devices was analyzed with a nontrivial solution (v_i^M , mode shapes) obtained from the govern equation. Randomly wrinkled OLED was modelled by controlling displacement boundary conditions in the pre-stretch state.³⁹ Then, the biaxial stretching of 3M elastomer was simulated by importing the results of the buckling analysis. c, Simulation model for each strain (0%, 30%, 65%, and 100%) to reflect the real situation when the device is mounted and stretched on the stretching jig. We have evaluated the mechanical simulation for the stress on various strains (0%, 30%, 65%, and 100%) of the GSOLED with 3M elastomer. We have reflected the Poisson' ratio in stretching model to overcome the limitation of low convergence and the restriction of stretching model.”

(Page 23, Table S1 in the revised supplementary information)

Table S1. Comparison among the GSOLEDs reported in articles^{27,28} and in this work.

Features	ACS Appl. Mater. Interfaces 2016, 8, 31166–31171 [27]	Organic Electronics 2017, 48, 314 [28]	This Work
Device Structure			Elastomer	3M VHB (~ 1000 μm)	PDMS	Thin Elastomer 100 μm
Encapsulation	X	X	SiN_x Passivated Film and Side NOA63 Encapsulation
Color Coordinate after Stretching	Shift	Shift	No Shift
Efficiency roll-off	O	O	X
Heat Dissipation	X	X	O
Current Efficiency	71cd/A, @205cd/m ²	X	(Thin elastomer or SiO₂ NPs) 99.4cd/A, @20,000cd/m²
Geometrical Stretchability	50%	3%	30% (Thin elastomer) 100% (3M elastomer)
Pre-Strain	Mechanically (100%)	Thermally (3%)	Mechanically (30%, 100%)
Mechanical Simulation	X	O	O
Optical Scattering	X	X	O

Reviewer #2

Recommendation: Major Revision

The authors demonstrated a highly efficient, heat-dissipating, geometrically twistable, stretchable organic light-emitting diode (GTS OLED) based on thin oxide/metal/oxide electrodes, pre-strained elastomer, and thin ultra-violet curable polymer encapsulation layer with silicon dioxide nanoparticles. Here, to have stretchability, the authors used pre-strained 3M elastomer to make random wavy buckles. A thin oxide-metal-oxide electrode and thin ultra-violet curable polymer can be under strain based on random wavy buckles even though they have high young modulus. Using the transparent oxide/metal/oxide electrodes shows uniform emission without chromic shift under stretching, and using the silicon dioxide nanoparticles in the encapsulation layer achieves heat-dissipating properties.

This work builds on previously well-developed technologies around geometrically stretchable electronics, which means general ideas used for geometrically stretchable organic light-emitting diodes are not completely new. Also, the ultra-violet curable polymer, especially for a Norland Optical Adhesive 63, has been used for encapsulation in stretchable OLEDs (ref. 18-20). The silicon dioxide nanoparticles are used to added to the NOA63 film to help light scattering to enhance the outcoupling efficiency (ref. 32). However, the authors systematically integrated the mentioned technologies and fabricated GTS OLEDs with enhanced electrical performance, and heat dissipation, stretchability, and uniform emission without chromic shift. The authors also provide the mechanism and dynamics of fabricated GTS OLEDs based on oxide/metal/oxide electrodes. I think the application level and stable electrical performance as a contribution to the field of stretchable electronics may draw some interest of readers of Nature Communications, but I do not think the current version reaches the appropriate level for publication -- major revisions in re-organizing the content and adding more detailed discussion according to the major comments and questions below are required.

Major comments and questions

Comment 1: The authors should address heat dissipation technologies for the organic light-emitting diodes and flexible/stretchable electronics in the introduction. In addition, the manuscript should include additional details and reviews on oxide/metal/oxide electrodes.

Authors Reply: Thank you for your valuable comments. We have addressed heat dissipation technologies for organic light-emitting diodes and flexible/stretchable electronics in the introduction. Transparent electrodes based on dielectric/metal/dielectric and an oxide-metal-oxide (OMO) is a promising candidate that can be tuned to produce highly conductive and transparent multilayer electrode exploiting the conductivity of the metal, the optical interference within the multilayers and the surface plasmonic effect at the metal/oxide interface.

In terms of flexibility and stretchability, a thin and ductile metal layer has the role of resisting fracture since it acts as the crack stopper or retardation on an elastomeric substrate. [RSC Adv., 2015, 5, 65094] Also, we have briefly reviewed MAM in the introduction and added cycle test of

the MAM electrode for stability in the result section. Additionally, we have compared three electrodes - PEDOT:PSS, ITO, and MAM – in the supplementary Figure S2.

(Page 5, line 12 ~ 21 in the revised manuscript)

“However, heat generation during operation may cause degradation of an OLEDs,^{29, 30} such as exciton quenching from triplet–triplet annihilation (TTA) and triplet-polaron annihilation (TPA) at high exciton densities in thick substrate.^{31, 32, 33} There are two strategies to dissipate the generated heat from the OLEDs; one is the “heat sink” that requires an additional thermally conductive layer in the device and the other one is the “heat dissipation” through thin substrate and encapsulation layer that does not require an extra conductive layer.³⁴ Especially, in OLEDs, a short heat transfer pathway can further facilitate heat dissipation without using a heat sink.²⁹ This pathway can be provided by incorporation of metal oxide nanoparticles (NPs) in the thin substrate and can enhance the heat dissipation in optoelectronic devices due to their high, thermal conductivity and porous scaffold design.³⁴”

(Page 4, line 13 ~ Page 5, line 2 in the revised manuscript)

“Alternatively, transparent electrodes based on dielectric/metal/dielectric and an oxide-metal-oxide (OMO), where a thin metal layer is sandwiched between two dielectric or oxide layers, is a promising candidate that can be tuned to produce highly conductive and transparent multilayer electrode exploiting the conductivity of the metal, the optical interference within the multilayers and the surface plasmonic effect at the metal/oxide interface.^{15, 16, 17, 18, 19} In terms of flexibility and stretchability, a thin and ductile metal layer has the role of resisting fracture since it acts as the crack stopper or retardation on an elastomeric substrate.¹⁹ Therefore, a thin OMO electrode could be proposed for stretchable electronics because of its better hole injection and optical transmittance, despite the thick oxide material being known for brittleness under high strains.²⁰ Among the OMOs, molybdenum trioxide (MoO₃)/gold (Au)/MoO₃ (MAM) has been found to be one of the promising candidates for the highly conductive and transparent electrodes.^{15, 21}”

(Page 4, line 13 ~ 17 in the revised supplementary information)

“e, In the MAM aspect, Hong et. al.⁸ reported that by optimizing the thickness of the outer oxide layer, the surface plasmon generated by the sandwiched metal could be coupled with the air using Bragg’s scattering, thus increasing the overall transmittance. f-h, The thickness of the sandwiched metal layer is also crucial as it causes a trade-off between transmittance and conductivity as reported by Wrzesniewski et al.⁹”

(Page 5, Supplementary Figure S2 in the revised supplementary information)

Transparent Electrodes	Advantages	Disadvantages	Ref
PEDOT:PSS	 High transmittance (>80%) Solution processable Well matched for Hole injection material Suitable for stretchable electronics (flexible) 	 Poor environmental stability ITO etched by acid PEDOT:PSS Not low sheet resistance 	[12]
ITO	 High transmittance (>90%) Low sheet resistance (<20 Ohm/sq) Well matched for Hole injection electrode 	 Poor mechanical flexibility thus not suitable for stretchable electronics High deposition temperature (>300 °C) Increasing cost of indium 	[13]
AgNWs	 Suitable for intrinsically, geometrically stretchable electronics High transmittance (>80%) Low sheet resistance (10 to 50 Ohm/sq) Good mechanical flexibility 	 Rough surface Poor environmental stability (native oxide) Network meltdown into nanoparticles 	[14]
MAM [MoO ₃ (15nm) / Au (14nm) / MoO ₃ (5nm)]	 Low deposition temperature (<300 °C) High transmittance (>80%) Low sheet resistance (10 to 50 Ohm/sq) Good mechanical flexibility High environmental stability Suitable for geometrically stretchable devices High work function (~5.25 eV) 	 Not suitable for intrinsically stretchable, suitable geometrically stretchable electronics 	[3]

Comment 2: The authors mentioned that geometrically stretchable OLEDs (GSOLEDs) were previously introduced, and this research realized geometrically twistable, stretchable OLEDs (GTS OLEDs). What is the difference between previously GSOLEDs and these GTS OLEDs?

Authors Reply: Thank you for your helpful comments. To avoid any confusion, we have unified the abbreviation to “twistable GSOLED” throughout the revised manuscript when we applied twist mode. The main difference is that our device is twistable (1D twist mode in Figure 2c and video clip) without damage. Also, we have conducted cyclic test of twist mode and there was no severe change in characteristics during 100 cycles (1 cycle: 0° - 180° - 0°).

(Page 39, Figure 5 in the revised manuscript)

(Page 2, line 22 in the revised manuscript)

“**Keywords:** Twistable GSOLEDs (Geometrically Stretchable Organic Light-Emitting Diodes)”

(Page 6, line 1 ~ 3 in the revised manuscript)

“Various types of stretchable OLEDs including GSOLEDs (geometrically stretchable OLEDs, transferred to the pre-strained substrate, buckled after strain release)^{26, 27, 28, 35} and ISOLEDs (intrinsically stretchable OLEDs directly stretched)⁵ were previously introduced.”

(Page 6, line 13 ~ 14 in the revised manuscript)

“This research realized twistable GSOLEDs utilizing the superior optoelectronic characteristics of MAM electrode.”

(Page 12, line 9 ~ 12 in the revised manuscript)

“Moreover, the device performance at various twisting angles (0°, 90°, 180°) have been measured and the GSOLEDs were unaffected by the twisting. And, the twisting cyclic test did not show any changes in the device performance up to 100 cycles as shown in Fig. 5.”

Comment 3: There are two full names of GTS OLED, one is geometrically twistable, stretchable organic light-emitting diodes (Line 38), and the other one is geometrically twistable, stable, stretchable organic light-emitting diodes (Line 90). It would be nice to unify as one.

Authors Reply: Thank you for your kind comments. We have unified the abbreviation from “GTS OLED” to “twistable GSOLED” as we mentioned in our above response (comment # 2).

(Page 2, line 22 in the revised manuscript)

“Keywords: Twistable GSOLEDs (Geometrically Stretchable Organic Light-Emitting Diodes)”

(Page 6, line 1 ~ 3 in the revised manuscript)

“Various types of stretchable OLEDs including GSOLEDs (geometrically stretchable OLEDs, transferred to the pre-strained substrate, buckled after strain release)^{26, 27, 28, 35} and ISOLEDs (intrinsically stretchable OLEDs directly stretched)⁵ were previously introduced.”

(Page 6, line 13 ~ 14 in the revised manuscript)

“This research realized twistable GSOLEDs utilizing the superior optoelectronic characteristics of MAM electrode.”

(Page 12, line 9 ~ 12 in the revised manuscript)

“Moreover, the device performance at various twisting angles (0°, 90°, 180°) have been measured and the GSOLEDs were unaffected by the twisting. And, the twisting cyclic test did not show any changes in the device performance up to 100 cycles as shown in Fig. 5.”

Comment 4: The authors show the images of the device performance during twisting and stretching tests and water immersion in Figure 2. The camera images confirm that the devices work well under various operation modes, but there is no numerical data for how much performance degradation occurs when twisting or stretching and how well restoration is performed. The data at various applied 2D strains (0-100%) are in Table 1 and Figure 3, but there is no data for twisting mode. Furthermore, the label's uniformity attached to Figure 2a-c, (a: Camera Images, Stretching Images, Operation Test) should be improved.

Authors Reply: We appreciate your kind suggestion. We have included the data of the performance during stretching up to 100 cycles at an applied strain of 30%, demonstrating a high mechanical robustness, as shown in Table1 in the revised manuscript and Figure S6f in the supplementary information. For performance during twisting, we have included device efficiency data to account for how much performance degradation occurs after twisting up to 100 cycles. Accordingly, we have updated data for twisting (0, 90 and 180°) mode in Figure 5. There was no

severe change in the device performance during 100 cycles of both twisting and stretching. Also, we have added labels in Figure 2a (Camera Images). Finally, we have updated the manuscript according to the above changes.

Additionally, we have conducted water immersion test (LT0) for various encapsulations and substrates as shown below.

- **Device A:** NOA63 / 3M tape / NOA63
- **Device B:** NOA63 / 3M tape + NOA63 side passivation / NOA63
- **Device C:** NOA63 / SiN_x / 3M tape + NOA63 side passivation / SiN_x / NOA63 / SiN_x
- **Device D:** NOA63 / SiN_x / 3M tape + epoxy side passivation / SiN_x / NOA63 / SiN_x

The LT0 measurement was performed at a constant temperature and humidity (25 °C and 25%), respectively until the luminance of the devices dropped to 0% of the initial luminance (at 1,000 cd/m²) in water immersion condition. Results suggest that side passivation is critical to the device stability and the device lifetime in water immersion test. When an additional SiN_x thin film adapted as passivation (**C**), it showed LT0 almost twice compared with that of only NOA63 film (**B**). The device remained intact when epoxy (UV RESIN XNR 5570B1, Nagase ChemteX Corporation) side passivation (**D**) was applied. Thus, in the harsh condition like water immersion, the most important factor is the side encapsulation. Even though the side encapsulation with epoxy was quite effective, there is a trade-off between stretchability and encapsulation. So, it needs further study to improve the stretchable encapsulation for GSOLEDs.

Also, we have performed the device lifetime (LT50, Fig. 4e) under stretching condition (0, 30, 65 and 100% at 10,000 cd/m²). We found that there was no severe change with various strain condition.

(Page 11, line 15 ~ Page 12, line 9 in the revised manuscript)

“As suggested in the Figure 2, the device lifetime (LT) under various encapsulation schemes (Device A : NOA63/only 3M tape side passivation/NOA63; Device B : NOA63/3M tape and NOA63 side passivation/NOA63; Device C : NOA63/SiN_x/3M tape and NOA63 side passivation/SiN_x/NOA63/SiN_x; Device D : NOA63/SiN_x/3M tape and epoxy side passivation/SiN_x/NOA63/SiN_x) and water immersion condition have improved and were comparable, respectively (shown in Fig. 4c, d). All of data trend was found to be coherent with the conditions. It may be concluded that the “side passivation” from a combination of 3M tape and NOA63 is quite critical as well as protective against “film penetration” through NOA63 films in a vertical direction. Device A encapsulation withstood for about 7 min during the water immersion, while Device B and C with NOA63 side passivation survived for over 1~3 hours. The shorter LT0 of the device A encapsulation as compared to its LT50 (Fig. 4c, d) is due to side penetration of water, which was confirmed by WVTR measurement (Supplementary Figure S3). Even though the side encapsulation with epoxy (Device D) was quite effective (Fig. 4d), there is a trade-off between stretchability and encapsulation. It needs further study to improve the stretchable encapsulation for GSOLEDs. Additionally, the LT50 at the different strain conditions

(0%, 30%, 65%, 100%) was also analyzed with 3M elastomer and the tendency remained the same despite of the strain. It means that geometrical stretching does not affect the device operation due to the mechanical consideration.”

(Page 11, Supplementary Figure S6f in the revised supplementary information)

(Page 38, Figure 4 in the revised manuscript)

(Page 36, Figure 2 in the revised manuscript)

(Page 39, Figure 5 in the revised manuscript)

Comment 5: In Figure 2, there is no notification about the samples that incorporate NOA63 film with NPs or without NPs both in text and figure caption.

Authors Reply: We appreciate your concern about missing information of NPs. We mentioned the NPs in the figure label or in the text only when we used NPs. That means we did not use NPs in the devices mentioned in Figure 2. However, we have updated the caption for avoiding confusion accordingly.

(Page 32, line 12 ~ 13 in the revised manuscript)

“The NPs were not incorporated in these devices.”

Comment 6: Figure 3 does not clearly show the device performance with and without SiO₂ NPs and the heat sink mechanism. I notice that the Scheme in Figure 3a is the full device structure of GTS OLEDs. The authors show no critical efficiency roll-off issue through Figure 3c-d, even when the 3M elastomer does not allow efficient heat dissipation. Figures 3b, e-f are designed to show the effects of thin NOA63 film with NPs by comparing glass-based, NOA63 film only, and NOA63 film with NPs based on non-GTS devices without thick 3M elastomer.

First, the meaning of the labels 'heat accumulation' and 'heat sink or dissipation' in Figures 3a, b are not clear.

Second, in figure 3c-d, the readers will be unable to appreciate the heat sink effects of using NOA63 film with or without NPs rather than other encapsulation layers.

Third, Figure 3b, e-f and 4 compares the heat dissipating properties of thin NOA63 film and thick glass substrates. How about the heat dissipating properties when the devices use thin glass substrates or other thin, flexible, and stretchable substrates?

Fourth, the authors said that Ir(ppy)₃ was used as a dopant, but to achieve a high device efficiency Ir(ppy)₂ tmd is also used and analyzed. Why is there a difference between using Ir(ppy)₂ tmd for Figure 3a and Ir(ppy)₃ for Figure 3b?

Authors Reply: Thank you for your thoughtful comments. We have tried to define our device structure based on stretchability and heat dissipation property. We have fabricated the GSOLEDs with thin elastomer (100 μm, DOW Corning HTV processed by Apple Silicon) and expecting GS & heat dissipation at the same time. Initially, we have compared two kinds of device platforms. The first one was the “Stretchable platform” on thick 3M elastomer (1,000 μm) (without heat dissipation function) and the other one was the “Heat dissipation platform” only on NOA63 substrate without elastomer (without stretchable function). During this revision, we have

combined previous two platforms, which means “*Stretchable & Heat dissipation platform*” on thin elastomer (100 μm) with stretchability and heat dissipation functions. Below, we have responded to your four kinds of comments, separately.

First, the meaning of the labels 'heat accumulation' and 'heat sink or dissipation' in Figures 3a, b are not clear.

Authors Reply: We are appreciated for your concern. In the previous version, there was confusion in device label. We unified the expression both of ‘heat accumulation’ and ‘heat sink’ as one, ‘*heat dissipation*’. Initially, stretchable device with 3M elastomer was expressed as ‘heat accumulation’ and only NOA63 film device was ‘heat sink or dissipation’. However, we have redefined the two platforms as “*Stretchable & Non-Heat dissipation*” and “*Non-Stretchable & Heat dissipation*”. Additionally, we have integrated two platforms into one as “*Stretchable & Heat dissipation*”.

(Page 37, Figure 3a, b and Page 38, Figure 4a in the revised manuscript)

(Page 9, line 1 ~ 4 in the revised manuscript)

“The device performances have been compared between ‘GS devices’ with thick 3M elastomer and ‘non-GS devices’ without 3M elastomer. This differentiated between ‘stretchable & non-heat dissipating devices’ and ‘non-stretchable & heat dissipating devices’^{29, 30, 31, 32, 33} (Fig. 3a, b). The GS devices with 3M elastomer could not dissipate heat efficiently”

Second, in figure 3c-d, the readers will be unable to appreciate the heat sink effects of using NOA63 film with or without NPs rather than other encapsulation layers.

Authors Reply: Thank you for your valuable comments. As you pointed out, we could not expect heat dissipation in Fig. 3c, d because of the thick 3M elastomer as we mentioned earlier. However, we have measured time resolved EL (TREL) of NOA63 devices with NPs to check how NPs work for heat dissipation in the device. We have found that NPs dissipated the heat efficiently due to their highly, thermally conductive (1.3 W/m K) SiO₂ NPs and porous scaffold design [Energy & Environmental Sci., 13, 5059-5067 (2020)], compared to that of NOA63 (0.2 W/m K). Although the thickness of the NOA63 containing NPs (16.3 µm) was around twice that of the film without NPs (9.2 µm), their identical TREL results (Fig. 6e) indicated that the effect of thicker film was minimized by the presence of NPs in dissipating heat. Additionally, we have updated the manuscript reflecting our response.

(Page 13, line 19 ~ Page 14, line 5 in the revised manuscript)

“On the other hands, although the thickness of the NOA63 containing NPs (NPs 4.5 wt%, 16.3 µm thickness) was twice that of the film without NPs (NOA63 9.2 µm), their identical TREL results (Fig. S18c) indicate that the effect of thicker film in dissipating heat was minimized by the presence of NPs. It was reported that Aluminum Oxide (Al₂O₃) NPs dissipated heat

efficiently from perovskite solar cells due to their highly, thermally conductive ($20 \sim 30 \text{ W m}^{-1} \text{ K}^{-1}$) metal oxide NPs and porous scaffold design (the schematic illustration of Fig. 6e inset).³⁴ Since the thermal conductivity of SiO_2 NPs ($1.3 \text{ W m}^{-1} \text{ K}^{-1}$)⁵⁵ is much higher than NOA63 ($0.2 \text{ W m}^{-1} \text{ K}^{-1}$), it can be inferred that SiO_2 NPs also facilitate heat dissipation from the device in a similar manner.”

(Page 40, Figure 6e in the revised manuscript and Page 33, Supplementary Figure S18b in the revised supplementary information)

Third, Figure 3b, e-f and 4 compares the heat dissipating properties of thin NOA63 film and thick glass substrates. How about the heat dissipating properties when the devices use thin glass substrates or other thin, flexible, and stretchable substrates?

Authors Reply: Thank you for your kind suggestion. Following your comments, we have measured the TREL of the device with thin elastomer (100 μm), which appeared between those with 70 μm and 210 μm NOA63 based devices without elastomer. It means that the thin elastomer efficiently dissipated the heat from the inside of the device. Consequently, the total thickness of the substrate (polymer film and elastomer) was dominant in the heat dissipation mechanism. For the cross-evaluation, we have fabricated the GS OLED with thin elastomer (100 μm) instead of 3M elastomer (1,000 μm). We have concluded that GS OLED with thin elastomer was not as good as only thin NOA63 (9.2 μm) device at low exciton density (below 10,000 nit)

due to the thickness of the thin elastomer, but they performed similarly over 10,000 nits. It means that thin elastomer devices show heat dissipation promptly.

(Page 40, Figure 6e in the revised manuscript and Page 33, Supplementary Figure S18b in the revised supplementary information)

(Page 14, line 5 ~ 9 in the revised manuscript)

“In addition, the TREL of the device with thin elastomer (100 μm) appeared between those with 70 μm and 210 μm NOA63 based devices without elastomer, which means that the thin elastomer efficiently dissipated the heat from the inside of the device. Consequently, the total thickness of the substrate (polymer film and elastomer) was dominant in the heat dissipation mechanism.”

(Page 37, Figure 3a, b and Page 38, Figure 4a in the revised manuscript)

(Page 11, line 1 ~ 14 in the revised manuscript)

“As we have discussed above, the stretchable devices could not show heat dissipation and heat dissipating devices could not be stretchable. To combine both qualities, additional devices have been fabricated using a thin elastomer (100 μm) at the different strain conditions (0, 10, 20 and 30%) as shown in Fig. 4. The heat dissipation function of the devices based on the thin elastomer were much efficient than that based on thick 3M elastomer, but less efficient than the NOA63 substrate only. Also, it was not as good as that of the device made only on thin NOA63 (9.2 μm) layers at low exciton density (below 10,000 nit) due to the thickness of the elastomer, but they performed similarly over 10,000 nits. The increment of CE with thin elastomer was a bit lower than that of only thin NOA63 substrate due to thin elastomer thickness (100 μm), which means that small quantity of heat still did not dissipate from thin elastomer. Moreover, the stretchability of thin elastomer ($\sim 30\%$) is much lower than that of 3M elastomer ($\sim 100\%$) because the thin elastomer could not sustain more than 30% pre-strain. We have also tried to fabricate stretchable and heat dissipating devices with even thinner elastomer (less than 100 μm); however, it was hard to be fabricated due to the handling difficulties of being torn over 30% strain.”

Fourth, the authors said that Ir(ppy)3 was used as a dopant, but to achieve a high device efficiency Ir(ppy)2 tmd is also used and analyzed. Why is there a difference between using Ir(ppy)2 tmd for Figure 3a and Ir(ppy)3 for Figure 3b?

Authors Reply: Thank you for your meaningful comments. Initially, we have used Ir(ppy)₃, a phosphorescent isotropic emitter, as a dopant with host CBP. Since horizontally oriented dipoles results in higher outcoupling efficiency than isotropic orientation of dipoles in OLEDs, we have used a phosphorescent horizontal emitter Ir(ppy)₂tmd with the same host for the high efficiency [Adv. Mater. 26, 3844-3847 (2014) & Chem. Mater. 27, 2767-2769 (2015)]. However, to avoid any confusion, we have used the same dopant Ir(ppy)₂tmd and updated the Figure 3b accordingly.

(Page 9, line 15 ~ 21 in the revised manuscript)

“In GSOLEDs, an phosphorescent isotropic emitter tris(2-phenylpyridine)iridium(III) [Ir(ppy)₃] was used as a dopant with an N,N-dicarbazolyl-4-4-biphenyl (CBP) host to optimize highly efficient GSOLEDs. Since horizontally oriented dipoles results in higher out-coupling efficiency than isotropic orientation of dipoles in OLEDs,⁵⁰ a phosphorescent horizontal emitter (bis(2-phenylpyridine)iridium(III) (2,2,6,6-tetramethylheptane-3,5-diketonate) [Ir(ppy)₂tmd] with the same host was analyzed and compared for high efficiency^{50, 51}”

(Page 37, Figure 3 in the revised manuscript)

Comment 7: At a 50% stretching strain of 3M tape, the OLED devices' overall strain level was approximately 3%. How much the overall strain level of OLED devices under a 100% stretching strain of 3M tape, and is it below the critical level of materials in OLED devices?

Authors Reply: We appreciate your comments. We have re-evaluated the mechanical simulation for the stress on various strains (0, 30, 65 and 100%) of the GSOLED with 3M elastomer. Previous method was limited to 50% strain due to the low convergence and the restriction of stretching model. During the revision, we have modified boundary conditions in gripping edges of substrate resulting in successful simulation until 100% stretching. We concluded that the device with low strain values (< 9%) worked well during stretching due to the wrinkled structure and stretchable elastomeric substrate. The device sustained the mechanical strain efficiently by adhesion of the sticky elastomer substrate. In general, it is widely known that thin films on compliant substrates can withstand larger strains than their intrinsic elongation at fracture without cracking. [Adv. Electron. Mater. 2018, 4, 1700429; Eng. Fract. Mech. 2002, 69 597–603 and Chem. Mater. 2019, 31, 9057-9069]

(Page 17, line 7 ~ 16 in the revised manuscript)

“The low stress in the pixel part was due to the wrinkled structure, formed intensively in the pixels through the pre-strain method, that could effectively release the mechanical strain of the device during the application of the tensile stress. Like the simulation results, the high mechanical reliability of the wrinkled structure in the device was also confirmed by various stability tests. Despite of the strain value of the device (< 9 %) in simulation, the actual device was not damaged in the stability tests since crack initiation and propagation was prevented by the adhesion of the sticky elastomer substrate^{60, 61, 62} (Fig. 8). The simulation result suggests that the geometrical wrinkled structure is suitable to fabricate stretchable OLEDs. Also, the experimental results on cycle tests (Supplementary Figure S6f) support the simulation result.”

(Page 43, line 10 ~ 15 in the revised supplementary information)

“c, Simulation model for each strain (0%, 30%, 65%, and 100%) to reflect the real situation when the device is mounted and stretched on the stretching jig. We have evaluated the mechanical simulation for the stress on various strains (0%, 30%, 65%, and 100%) of the GSOLED with 3M elastomer and reflected boundary conditions of gripping edges as in actual experiment for stretching model to overcome the limitation of low convergence and the restriction of stretching model.”

(Page 42, Figure 8 in the revised manuscript)

(Page 43, Supplementary Figure S24 in the revised supplementary information)

Comment 8: The full information of reference 14 is missing.

Authors Reply: Thank you for pointing out the error. we have fixed the reference.

Previous manuscript

14. Mukhopadhyay, R. (ACS Publications, 2007).

Revised manuscript

23. Mukhopadhyay R. When PDMS isn't the best. Anal Chem 79, 3248-3253 (2007)

(Page 28, line 33 in the revised manuscript)

“23. Mukhopadhyay R. When PDMS isn't the best. Anal Chem 79, 3248-3253 (2007)”

Minor comment on typographic errors

Comment 1: Check Ref. 41, 50.

Authors Reply: We appreciate your concern. Ref 41 (Ref 50 in the revised manuscript) explained horizontal emitter and Ref 50 (Ref 59 in the revised manuscript) mentioned VHB tape information. We have recognized that hyphen “-” made some typo error in the manuscript. So, we double-checked by conversion to pdf and there is no problem in the revised version.

Previous manuscript

- 41 Kim, K. H., Moon, C. K., Lee, J. H., Kim, S. Y. & Kim, J. J. Highly efficient organic light-emitting diodes with phosphorescent emitters having high quantum yield and horizontal orientation of transition dipole moments. *Advanced Materials* **26**, 3844-3847 (2014).
- 50 Ha, S. M., Yuan, W., Pei, Q., Pelrine, R. & Stanford, S. Interpenetrating polymer networks for high-performance electroelastomer artificial muscles. *Advanced Materials* **18**, 887-891 (2006).

Revised manuscript

50. Kim KH, Moon CK, Lee JH, Kim SY, Kim JJ. Highly Efficient Organic Light-Emitting Diodes with Phosphorescent Emitters Having High Quantum Yield and Horizontal Orientation of Transition Dipole Moments. *Advanced Materials* **26**, 3844-3847 (2014).
59. Ha SM, Yuan W, Pei QB, Pelrine R, Stanford S. Interpenetrating polymer networks for high-performance electroelastomer artificial muscles. *Advanced Materials* **18**, 887-891 (2006).

(Page 30, line 37 in the revised manuscript)

“50. Kim KH, Moon CK, Lee JH, Kim SY, Kim JJ. Highly Efficient Organic Light Emitting Diodes with Phosphorescent Emitters Having High Quantum Yield and Horizontal Orientation of Transition Dipole Moments. *Advanced Materials* 26, 3844-3847 (2014).”

(Page 31, line 25 in the revised manuscript)

“59. Ha SM, Yuan W, Pei QB, Pelrine R, Stanford S. Interpenetrating polymer networks for high-performance electroelastomer artificial muscles. *Advanced Materials* **18**, 887-891 (2006)”

Reviewer #3

Recommendation: Rejection

The authors reported geometrically stretchable OLED by the NOA63 substrates and MAM electrodes. The device showed 100% stretchability by being laminated onto 1mm thick elastomer, and stability in mechanical strain and water immersion. Furthermore, the authors investigated the effect of substrate thickness on the efficiency roll-off, and stability of EL intensity over strain with Ag and MAM electrodes. However, the paper shows poor novelty as research. For example, stretchable OLED using buckling is already demonstrated. The advantage of MAM over other transparent electrodes (e.g. PEDOT:PSS, ITO, AgNWs) are not shown. Although the reviewer evaluate the investigation on the effect of substrate thickness and microcavity effect, the novelty is not enough for Nature Communications. Besides, the authors do not cite papers properly. Therefore, the reviewer recommends the rejection of this manuscript from Nature Communications. The followings are the detailed comments.

Authors Reply: Thank you for your kind comments. Before answering your concerns one by one, we would like to summarize the novelty of our work in advance.

1) stretchable OLED using buckling is already demonstrated.

→ We agree with your opinion. However, we have clarified the actual device was not damaged in the stability tests since crack initiation and propagation was prevented by the adhesion of the sticky elastomer substrate despite of the strain value of the device (< 9 %) in simulation. Additionally, we have fabricated GSOLEDs with thin elastomer and found that there were two important factors for micro buckling formation; one is the thickness of substrate and the other one is the thickness of elastomer. (detailed explanation in comment 1)

2) The advantage of MAM over other transparent electrodes (e.g. PEDOT:PSS, ITO, AgNWs) are not shown.

→ As you mentioned, we did not describe the MAM electrode enough in the previous manuscript. For reinforcement, we have added the Fig. S2 in supplementary information showing that advantages of MAM against PEDOT:PSS, ITO and AgNWs. We have revised it in our manuscript accordingly. (detailed explanation in comment 2)

3) Although the reviewer evaluates the investigation on the effect of substrate thickness and microcavity effect, the novelty is not enough for Nature Communications.

→ We are absolutely agreeing with your opinions. The novelties were not clearly emphasized efficiently in previous manuscript. However, after revision, there are lots of updates acknowledging the valuable comments of the reviewers.

We have conducted two simulations: one is the mechanical simulation and the other one is the optical simulation. We found that thin NOA63 as a substrate dissipated heat efficiently and NOA63 with NPs was helpful for optical out-coupling with heat dissipation. Also, we have tried to fabricate devices with various encapsulation conditions and found that side encapsulation was

tremendously critical for device stability. However, there is a trade-off relation between stretchability and encapsulation. So, it needs further study to improve the encapsulation with stretchability for GSOLED.

Furthermore, we have compared two kinds of device platforms. The first one was the “Stretchable platform” on thick 3M elastomer (1,000 μm) (without heat dissipation function) and the other one was the “Heat dissipation platform” only on NOA63 substrate without elastomer (without stretchable function) as shown in Fig. 3a, b. During this revision, we have combined previous two platforms, which means “Stretchable & Heat dissipation platform” on thin elastomer (100 μm) with stretchability and heat dissipation functions as shown in Fig. 4a.

We have updated the manuscript comparing the TREL data of the devices with and without NPs to check the exciton recombination mechanism, again. When the glass substrate was adapted, heat could not dissipate from the devices. In case of thin NOA63 without 3M elastomer, heat could dissipate efficiently as shown in Fig. 6b. Therefore, we have concluded from the TREL result that the substrate thickness affects the heat dissipation of the devices critically.

In addition, it has been already reported that metal oxide NPs dissipated heat efficiently in optoelectronic devices due to their high thermal conductivity and porous scaffold design. [Energy & Environmental Sci., 13, 5059-5067 (2020)] Because of that, even though NOA63+NPs (16.3 μm) substrate was thicker than NOA63 (9.2 μm), almost similar TREL results (Fig. 6e) suggested that NPs facilitated the heat dissipation mechanism due to the thermally conductive NPs ($1.3 \text{ W m}^{-1} \text{ K}^{-1}$) compared to the NOA63 ($0.2 \text{ W m}^{-1} \text{ K}^{-1}$) and porous scaffold design. Therefore, it could be concluded that the SiO_2 NPs are beneficial for both optical out-coupling and the heat dissipation mechanism.

The summary of our work is as follows.

- I. *The device lifetime of GSOLEDs in various condition, such as encapsulation method, water immersion and strain condition.*
- II. *The GSOLEDs based on the MAM with the invariant color-coordinate.*
- III. *The relation in GSOLEDs between substrate thickness and heat dissipation thorough the TREL and suggestion of the exact mechanism as shown in Fig. 6.*
- IV. *The demonstration of the stretchable and heat dissipating device with thin elastomer (100 μm), instead of 3M elastomer (1,000 μm) simultaneously as shown in Fig. 4a, b. (To our knowledge, it has not been reported before)*
- V. *The application of the various encapsulation efficiently for GSOLED with enhanced WVTR (detailed explanation in comment 3)*
- VI. *The SiO_2 NPs are beneficial for both optical out-coupling and the heat dissipation mechanism, concurrently.*

We have added a significant amount of new data during this revision that, we believe, fully supports the novelty of our work. We hope that this revision could satisfy your expectations.

4) the authors do not cite papers properly

→ As you pointed out, we missed some citation for our manuscript. So, we have fixed suitably (detail explanation on comment 4)

(Page 37, Figure 3a, b and Page 38, Figure 4a in the revised manuscript)

(Page 40, Figure 6 in the revised manuscript)

Comment 1: As authors cite, geometrically stretchable OLED has been demonstrated in many other literatures. Besides, the substrate thickness of 9.2 μ m is very thick compared with the previous reports where they are using 2-3 μ m thick plastic as a substrate. This thick substrate thickness caused very large buckling periodicity (\sim 500 μ m), which makes it difficult in the application due to the bad appearance. NOA with SiO $_2$ is 16.3 μ m, which makes the appearance as “stretchable OLED” even worse.

Authors Reply: Thank you for your kind comments. We agree that 16.3 μ m-thick plastic substrate should cause larger buckling periodicity than the 2-3 μ m thick plastic substrate as reported earlier. [Org. Electron. 43, 77–81 (2017), using 12,000 rpm] Also, it was reported that the buckling periodicity depends on both polymer substrates and elastomer; that means, the overall thickness of the device including substrate and elastomer determines the periodicity of wrinkled structure. [Adv. Mater. 31, 1807516 (2019); Adv. Optical Mater. 8, 1901525 (2020) & Adv. Mater. Technol. 5, 200231 (2020)]

Following your comments, we have tried to form thin NOA63 film at high rpm, such as 6000, 7000 and 8000. But it was hard to get the thin NOA63 due to the equipment limitation (not allowed over 6000 rpm). Because of that, we have obtained uniform and stable film, 9.2 μ m-NOA63 film without NPs and 16.3 μ m-thick NOA63 film with NPs, respectively (with 5000 rpm). The average periodicity of random buckling of film on 3M elastomer was 300 ± 72 μ m

(NOA63) and $783 \pm 136 \mu\text{m}$ (NOA63+NPs) as shown in the results of the confocal image (Figure S5 in the supplementary information).

As a plan B, we have tried it with thin elastomer (100 μm) and found that the buckling periodicity of film on thin elastomer decreased to $155 \pm 53 \mu\text{m}$. Thus, we found that there were two important factors for micro buckling formation; one is the thickness of substrate and the other one is the thickness of elastomer.

(Page 10, line 12 ~ 17 in the revised supplementary information)

“d, More small buckling periodicity (average value, $\sim 155 \mu\text{m}$) occurred in thin elastomer (100 μm), while large buckling periodicity (average value, $\sim 300 \mu\text{m}$) in thick elastomer (1,000 μm) was observed. It was reported that the buckling periodicity depends on both polymer substrates and elastomer; that means, the overall thickness of total device including substrate and elastomer determines the periodicity of wrinkled structure.^{19, 20, 21} Thus, we found that there were two important factors for micro buckling formation; one is the thickness of substrate and the other one is the thickness of elastomer.”

(Page 9, Supplementary Figure S5 in the revised supplementary information)

Peak to Peak of buckle formation

Substrates	1 st	2 nd	3 rd	4 th	Avg.
3M Elastomer + NOA63	220 μm	260 μm	350 μm	370 μm	300 \pm 72 μm
3M Elastomer + NOA63 + NPs	590 μm	800 μm	830 μm	910 μm	783 \pm 136 μm
Thin Elastomer +NOA63	204 μm	102 μm	182 μm	132 μm	155 \pm 53 μm

- MAM : MoO₃ (15 nm) / Au (14 nm) / MoO₃ (5 nm) * Ag (20 nm)
- NOA 63 (9.2 μm) * 3M Elastomer (1,000 μm) * Thin Elastomer (100 μm)

Comment 2: Geometrically stretchable OLEDs using transparent electrodes (PEDOT:PSS, ITO, AgNWs) have been already demonstrated. The authors should clearly show the advantage of MAM over them.

Authors Reply: Thank you for your valuable comments. Various studies on stretchable and flexible electrodes has been conducted. Even though these approaches provide considerable enhancements, there is still room for further improvement in terms of device stability and efficiency. The PEDOT:PSS, AgNWs and ITO have their own drawbacks for GSOLED as referred in the revised manuscript.

On the other hand, MAM electrodes possess some excellent properties like high transmittance, high work function, high environmental stability, and good mechanical stability as suggested in Fig S2 in the revised supplementary information. In terms of flexibility and stretchability, a thin and ductile metal layer has the role of resisting fracture since it acts as the crack stopper or retardation on an elastomeric substrate. [RSC Adv., 2015, 5, 65094] We have updated our manuscript mentioning the advantages of MAM over other electrodes including PEDOT:PSS, ITO, and AgNWs promptly.

(Page 4, line 7 ~ 13 in the revised manuscript)

“This included a graphene-based electrode,¹⁰ a highly transparent metal-grid¹¹ and polymer with silver nanowires (AgNWs).^{12, 13} Even though these approaches provide considerable enhancements, there is still room for further improvement in terms of device stability and efficiency. The PEDOT:PSS⁶ and AgNWs electrodes^{12, 13} are environmentally unstable, AgNWs network has rough surface and meltdown problem,¹⁴ and ITO is not suitable for flexible and stretchable electronics since the crack onset strain of ITO is typically 1% or less.^{8, 9}”

(Page 5, Supplementary Figure S2 in the revised supplementary information)

Transparent Electrodes	Advantages	Disadvantages	Ref
PEDOT:PSS	 High transmittance (>80%) Solution processable Well matched for Hole injection material Suitable for stretchable electronics (flexible) 	 Poor environmental stability ITO etched by acid PEDOT:PSS Not low sheet resistance 	[12]
ITO	 High transmittance (>90%) Low sheet resistance (<20 Ohm/sq) Well matched for Hole injection electrode 	 Poor mechanical flexibility thus not suitable for stretchable electronics High deposition temperature (>300 °C) Increasing cost of indium 	[13]
AgNWs	 Suitable for intrinsically, geometrically stretchable electronics High transmittance (>80%) Low sheet resistance (10 to 50 Ohm/sq) Good mechanical flexibility 	 Rough surface Poor environmental stability (native oxide) Network meltdown into nanoparticles 	[14]
MAM [MoO ₃ (15nm) / Au (14nm) / MoO ₃ (5nm)]	 Low deposition temperature (<300 °C) High transmittance (>80%) Low sheet resistance (10 to 50 Ohm/sq) Good mechanical flexibility High environmental stability Suitable for geometrically stretchable devices High work function (~5.25 eV) 	 Not suitable for intrinsically stretchable, suitable geometrically stretchable electronics 	[3]

Comment 3: WVTR of NOA with 3M tape is reported. However, considering the thickness, it is not impressive.

Authors Reply: We appreciate your concern regarding the encapsulation property. According to your comments, we have tried to get better the water vapor transmission ratio (WVTR) value. We have employed various encapsulation process to improve it.

The WVTR by the calcium (Ca) test for various encapsulations and substrates was analyzed as shown below.

- **Device A:** NOA63 / 3M tape / NOA63
- **Device B:** NOA63 / 3M tape + NOA63 side passivation / NOA63
- **Device C:** NOA63 / SiN_x / 3M tape + NOA63 side passivation / SiN_x / NOA63 / SiN_x
- **Device D:** NOA63 / SiN_x / 3M tape + epoxy side passivation / SiN_x / NOA63 / SiN_x

The values of WVTR obtained for the samples are summarized in the table of Figure S3.

From the WVTR result, we speculate that the 3M tape alone as a sealant was not the proper material to use for the purpose, as it could have allowed moisture to penetrate the substrate. Comparing WVTR for **C** and **D**, epoxy (UV RESIN XNR 5570B1, Nagase ChemteX Corporation) was found to be better than NOA63 in blocking the side from the moisture. Even though encapsulation in **C** was not as good as that in **D**, the WVTR value of the sample **C** ($3.6 \times 10^{-2} \text{ g}\cdot\text{m}^{-2} \text{ day}^{-1}$) is comparable to that of 4-6 layers of recent graphene-encapsulation work ($1.78 \times 10^{-2} \text{ g}\cdot\text{m}^{-2} \text{ day}^{-1}$) [ACS Appl. Mater. Inter. 8, 14725-14731 (2016)]. Thus, it may be concluded that the side encapsulation is much more important than the penetration from top and bottom films.

Additionally, we have applied same encapsulation methods to the devices for the lifetime measurements as shown in Fig. 4c, d. The LT50 measurement was performed at a constant temperature and humidity (25 °C and 25%) until the luminance of the devices dropped to 50% of the initial luminance (1000 cd/m^2) and LT0 was similar with LT50 but measured until 0% luminance in the water condition. Even though the result of lifetime measurements suggested that the side encapsulation with epoxy was quite effective, there is a trade-off between stretchability and encapsulation. It needs further study to improve the encapsulation with stretchability for GSOLEDs.

(Page 6, Supplementary Figure S3 in the revised supplementary information)

$$\begin{aligned}
 \text{WVTR} &= \rho(\text{CaO}) \times \frac{m(\text{H}_2\text{O})}{m(\text{CaO})} \times \frac{D(\text{CaO})}{t} \quad [\text{g}/\text{m}^2 \text{ day}] \\
 &= 3.35 \times 10^6 \text{ g}/\text{m}^3 \times \frac{18.01528 \text{ g}/\text{mol}}{56.077 \text{ g}/\text{mol}} \times D(\text{CaO})/t \quad [\text{g}/\text{m}^2 \text{ day}] \\
 &= 1.07622 \times 10^6 \text{ g}/\text{m}^3 \times D(\text{CaO})/t \quad [\text{g}/\text{m}^2 \text{ day}] \\
 &\Rightarrow 1.07622 \times 10^6 \text{ g}/\text{m}^3 \times \left(\frac{T(\text{Final}) - T(\text{Initial})}{100 - T(\text{Initial})} \times 500 \times 10^{-10} \text{ m} \right) / \left(\text{Time}(\text{min}) \times \frac{1\text{h}}{60\text{min}} \times \frac{1\text{day}}{24\text{h}} \right) \\
 & * \rho: \text{Density}, m: \text{Molar concentration}, D: \text{Thickness}, t: \text{Time}, T: \text{Transmittance}
 \end{aligned}$$

(Page 38, Figure 4 in the revised manuscript)

Comment 4: The followings are the examples of “not properly cited” references. The reviewer strongly recommend to revise before submitting elsewhere. Page 1 “However, these methods have drawbacks including poor uniformity, process complexity, low conductivity, instability, low transmittance, and work function (ϕ) mismatches for carrier injection” has no reference or evidence to prove. Reference 5 has nothing to do with the hygroscopicity of polyurethane. Reference 14 is not cited properly. Ref 26 is not intrinsically stretchable OLED.

Authors Reply: Thank you for your kind and considerate comments. We really appreciate your points because we could fix our errors in the revised manuscript. As you told, we have checked the references and confirmed as bellow table.

Submitted manuscript	Revised manuscript	Contents	Changed
Ref 5	Ref 7	Thin ITO	stretchable electronics, ^{4,5} → (ITO) ^{7, 8, 9}
Ref 14	Ref 23	PDMS elastomer	-
Ref 26	Ref 35	GSOLELED, not ISOLED	(ISOLEDs, directly stretched) ^{5,26} → GSOLELEDs (transferred to the pre-strained substrate, buckled after strain release) ^{26, 27, 28, 35}

We have added references to the other electrodes which was updated in the revised manuscript. We have renewed the manuscript regarding Ref 5 related with ITO (Ref 7 in the revised manuscript) and 26 related to GSOLELED (Ref 35 in the revised manuscript) properly by replacing those references at appropriate locations. Also, Ref 14 related to PDMS (Ref 23 in the revised manuscript) has been cited properly in the manuscript.

(Page 4, line 5 ~ 13 in the revised manuscript)

“This interest has led to various studies on electrodes, as a replacement of poly(3,4-ethylenedioxythiophene):poly(styrene sulfonate) (PEDOT:PSS)⁶ and indium tin oxide (ITO)^{7, 8, 9}. This included a graphene-based electrode,¹⁰ a highly transparent metal-grid¹¹ and polyimide with silver nanowires (AgNWs).^{12, 13} Even though these approaches provide considerable enhancements, there is still room for further improvement in terms of device stability and efficiency.^{12, 13} The PEDOT:PSS⁶ and AgNWs electrodes are environmentally unstable, AgNWs network has rough surface and meltdown problem,¹⁴ and ITO is not suitable for flexible and stretchable electronics since the crack onset strain of ITO is typically 1% or less.^{8, 9}”

(Page 5, line 6 ~ 7 in the revised manuscript)

“However, the hygroscopicity of the material made it unreliable, when used for stretchable polymer LEDs.²⁴”

(Page 6, line 1 ~ 3 in the revised manuscript)

“Various types of stretchable OLEDs including GSOLELEDs (geometrically stretchable OLEDs, transferred to the pre-strained substrate, buckled after strain release)^{26, 27, 28, 35} and ISOLEDs (intrinsically stretchable OLEDs directly stretched)⁵ were previously introduced.”

(Page 28, line 33 in the revised manuscript)

“23. Mukhopadhyay R. When PDMS isn't the best. Anal Chem 79, 3248-3253 (2007)”

Our self-correction

We have updated some figures to correct typos and improve the integrity and visual appearance of the figures.

(Page 40, Fig. 6c in the revised manuscript)

Figure 6c (previously Figure 4c): We have fixed typos in the figure caption and in the label of the y-axis changing “Substrates tamp” to “Substrates temp.” and “Normalized Temp. ($\Delta T/T_0$)” to “Normalized Temp. (T/T_0)”, respectively.

(Page 41, Fig. 7l in the revised manuscript)

Figure 7l (previously Figure 5l): We have updated the images for the Ag- and MAM-based electrodes at various measurement angles because it was switched.

We have highlighted the revised sentences in **blue color**. With the incorporated changes, we hope the manuscript is now acceptable for publication in the **Nature Communications**. Should you have any further questions or comments, please do not hesitate to contact me.

Sincerely,
Seung Yoon Ryu
Division of Display & Semiconductor Physics,
College of Science and Technology,
Korea University (Sejong Campus),
(30019) 2511, Sejong-ro, Sejong,
Republic of Korea,

REVIEWER COMMENTS

Reviewer #1 (Remarks to the Author):

The authors achieved highly efficient/stretchable OLED by using MOM as a transparent electrode and ultra-violet-curable polymer, NOA63/3M elastomer, as substrate. Especially, by doping SiO₂ NPs into NOA63, which has good heat dissipation characteristics, and by reducing the reflectance of the substrate, the device efficiency could be improved. These devices exhibited a high current efficiency of 82.5 cd/A and EQE of 22.6% with minimum efficiency roll-off, which didn't show any change of CIE coordinate after stretching. Electrical and optical characterization are well organized and are in good agreement with experimental results. Therefore, the reviewer suggests the publication of this paper to 'Nature communications'.

Reviewer #2 (Remarks to the Author):

The authors have presented a complete set of revisions to address the reviewer comments. I feel that the revised version is suitable for publication in its current form.

Reviewer #3 (Remarks to the Author):

The authors conducted a lot of additional experiments and revision in text to improve the quality of the manuscript. The revised manuscript and communications with the reviewers gave me better understanding of the manuscript. However, the manuscript is still lack of novelty, and the manuscript is now too complicated to follow. The reviewer can recommend the publication of this manuscript in Nature Communications only if the authors address the following concerns properly.

1. The authors report smaller buckling can be achieved by reducing the thickness of NOA film or elastomer. However, as clearly observed in Fig. 2b and 4e, most of the mechanical strains are not observed in these small buckling. The device absorb strain not by buckles, but by flexing in very large diameter. In Fig. 4e. The size of light emitting area does not change as described by the strain. The author should define the strain by the change of light emitting area.

2. Comparison with the other transparent electrodes are not enough. Although the authors criticize many points of the other electrodes, there are no quantitative comparison. MoO_x/Au/MoO_x can be also criticized in same way. MoO_x is water soluble. Au is very expensive... If the author cannot demonstrate the clear advantage of MAM, the reviewer recommends the authors to change the title and stop criticizing the other transparent electrodes.

3. The reviewer conducted additional experiments to test the performance of encapsulations. Still, the importance of SiN_x and edge encapsulation is well known. In addition, stretchable OLED is not demonstrated with the best encapsulation.

4. In page 34 of the rebuttal letter, the authors summarize their work. Here the reviewer shows their opinion on each point.

> The device lifetime of GSOLEDs in various condition, such as encapsulation method, water immersion and strain condition.

As commented in this comment 3, it is difficult to find the novelty.

> The GSOLEDs based on the MAM with the invariant color-coordinate.

The invariant color coordinate happens only when nontransparent electrodes are used. It is not a general issue for all the stretchable OLEDs. In addition, the importance of the use of MAM is not demonstrated as the reviewer mentioned in Comment 2.

> The relation in GSOLEDs between substrate thickness and heat dissipation thorough the TREL and suggestion of the exact mechanism as shown in Fig. 6.

Heat dissipation by thin substrate maybe a major novelty of this paper, but the current manuscript

contains too many not-so-novel parts. The reviewer recommends to cut down these parts and revise the manuscript dramatically. In addition, the authors should compare the heat-dissipation performance with the other approaches.

> The demonstration of the stretchable and heat dissipating device with thin elastomer (100 μm), instead of 3M elastomer (1,000 μm) simultaneously as shown in Fig. 4a, b. (To our knowledge, it has not been reported before)

It may not be reported, but the definition of the stretchability of this device is doubtful as written in my comment 1.

> The application of the various encapsulation efficiently for GSOLED with enhanced WVTR (detailed explanation in comment 3)

This investigation is not new as described in comment 3. Much better water stability of flexible OLEDs has been shown elsewhere.

> The SiO₂ NPs are beneficial for both optical out-coupling and the heat dissipation mechanism, concurrently.

As pointed out by the other reviewer, this point is not new. as shown in the following paper, high optical out-coupling and higher EQE is already demonstrated just by reducing the substrate thickness. <https://doi.org/10.1016/j.orgel.2019.03.040>

5. Current manuscript has no focus in the manuscript and is too long. The manuscript is very tough to read even for reviewers. Nature communications attract lots of attention from researchers in various field. The reviewer strongly recommend to revise the manuscript dramatically.

March 8, 2021

Dear Reviewers,

Thank you very much for your response for the article “*Highly Efficient, Heat Dissipating, Stretchable Organic Light-Emitting Diodes Based on an $\text{MoO}_3/\text{Au}/\text{MoO}_3$ Electrode with Encapsulations*”. I am grateful to the kind reviewers for the comments that helped us improve the comprehension of our manuscript and provided us useful insight to enhance the work. Reviewer #1 and Reviewer #2 accepted our previous revision, whereas reviewer #3 did not agree with that in some points. We have taken the comments of reviewer #3 sincerely and carefully examined the stated comments. Therefore, all authors have tried our best to respond to those queries with supporting literature and analysis. Kindly, catch the following bulleted list of major changes that we have made in the revised manuscript and the supporting information. We hope that our responses to the queries/comments would provide a satisfying comprehension of the proposed work and believe our revised manuscript could satisfy all readers and referees. If any further improvements would be required, we shall try our best to address the issues. Again, we are deeply grateful for considering our manuscript and looking forward to hearing the positive feedback from you soon.

- We have updated the title replacing “Oxide-Metal-Oxide Electrode” with “ $\text{MoO}_3/\text{Au}/\text{MoO}_3$ Electrode.”

- We have reduced the number of main figures by transferring those to supplementary figures and modified Table 1 to emphasize our own compact and solid novelties.
- We have compared the GSOLEDs with thick 3M elastomer and thin elastomer for validating the small and large buckling between pixel and non-pixel area.
- We have revised the introduction part of the manuscript for avoiding criticizing the other electrodes and updated the Fig. S2 accordingly, to reveal pros and cons of each electrode including our proposed electrode.
- We have added a comparison table (Table S2) for different techniques of heat sink or dissipation performance with the previously published works.

Sincerely,
Seung Yoon Ryu

**Reviewer #1**

**Recommendation: Accept**

The authors achieved highly efficient/stretchable OLED by using MOM as a transparent
electrode and ultra-violet-curable polymer, NOA63/3M elastomer, as substrate. Especially, by
doping SiO₂ NPs into NOA63, which has good heat dissipation characteristics, and by reducing
the reflectance of the substrate, the device efficiency could be improved. These devices exhibited
a high current efficiency of 82.5 cd/A and EQE of 22.6% with minimum efficiency roll-off,
which didn't show any change of CIE coordinate after stretching. Electrical and optical
characterization are well organized and are in good agreement with experimental results.

Therefore, the reviewer suggests the publication of this paper to 'Nature communications'.

**Reviewer #2**

**Recommendation: Accept**

The authors have presented a complete set of revisions to address the reviewer comments. I feel
that the revised version is suitable for publication in its current form.

**Reviewer #3**

**Recommendation: Major Revision**

The authors conducted a lot of additional experiments and revision in text to improve the quality
of the manuscript. The revised manuscript and communications with the reviewers gave me
better understanding of the manuscript. However, the manuscript is still lack of novelty, and the
manuscript is now too complicated to follow. The reviewer can recommend the publication of
this manuscript in Nature Communications only if the authors address the following concerns
properly.

**Comment 1:** The authors report smaller buckling can be achieved by reducing the thickness of
NOA film or elastomer. However, as clearly observed in Fig. 2b and 4e, most of the mechanical
strains are not observed in these small buckling. The device absorb strain not by buckles, but by

flexing in very large diameter. In Fig. 4e. The size of light emitting area does not change as
described by the strain. The author should define the strain by the change of light emitting area.

**Authors Reply:** Thank you for your valuable comments. As you pointed out, the large buckling
is dominant on the GSOLEDs in our work. Since small buckling is hard to be checked on the
operating devices, we have compared the buckling of non-operating devices with both thick 3M
elastomer and thin elastomer using confocal microscope. We have found that the average
buckling periodicities on the pixel area and non-pixel area are almost similar. Thus, the strain on
the devices affected pixel and non-pixel area in the same way.

Additionally, thinner elastomer is helpful to make smaller buckling periodicities.
(Supplementary Information Figure S5 and S7). Moreover, we have already defined the strain
based on light emitting area (pixel area) with thick 3M elastomer and newly added the strain
based on light emitting area (pixel area) with thin elastomer. For the complete comparison, we
have added the schematic illustration under the strain and table for the calculated strain based on
the buckled pixel area. (Supplementary Information Figure S7) When we released the strain from
the elastomer, the pixel areas (5 mm × 5 mm) shrunk to 3.5 mm × 3.5 mm (3M elastomer) and
4.4 mm × 4.4 mm (thin elastomer) according to the initial sizes of the elastomers. Thus, the 2D
strain in the GSOLEDs can be defined as percentage of the initially deformed pixel area (A_i) and
deformed pixel area (A_d) with respect to the non-deformed pixel area ($A_{nd} = 25 \text{ mm}^2$) that is,
strain = $[(A_{nd} - A_d) / A_i] \times 100\%$. i.e., 65 % strain = $[(25 \text{ mm}^2 - 16 \text{ mm}^2) / 12.25 \text{ mm}^2] \times 100\%$ at
the 3M elastomer.

Moreover, all buckling images in previous Fig. 2b and 4e were fabricated with thick 3M
elastomer, not with thin elastomer as it was hard to handle thinner substrate which demonstrated
a degradation over 30% strain. However, we have presented the possibility of reducing the
buckle size and improving the stretchability if more robust material with over 30% strain
capabilities will be realized. Therefore, in this work it was reasonable for large buckling to be
presented in previous Fig. 2b and 4e. We are sorry to provide you the misunderstanding on this.

**(Page 8, line 20 ~ Page 9, line 2 in the revised manuscript)**

“The fabricated wavy buckles were observed by optical microscope (OM) and confocal
microscope (CM) on the NOA63, with and without NPs. We have also measured the pixel area
and non-pixel area with CM for verifying buckling on the different parts of the devices
(Supplementary Figure S5 and S6). The buckling of thin elastomer showed smaller buckling
compared to thick 3M elastomer. Additionally, we have defined the strain based on light emitting
area (pixel area) with both types of elastomers (Supplementary Figure S7).”

**(Page 10, Supplementary Figure S5 in the revised supplementary information)**

Peak to Peak of Buckle Formation

Part		1 st	2 nd	3 rd	4 th	Avg.
3M Elastomer (1000µm)	Pixel Area	216 µm	320 µm	293 µm	312 µm	280±47 µm
	Non-Pixel Area	221 µm	322 µm	345 µm	349 µm	309±60 µm
Thin Elastomer (100µm)	Pixel Area	123 µm	144 µm	237 µm	121 µm	156±47 µm
	Non-Pixel Area	135 µm	105 µm	105 µm	140 µm	121±19 µm

(Page 10, line 2 ~ 8 in the revised supplementary information)

“Figure S5. Camera and confocal microscopy (CM) images of pixel and non-pixel area in
the GSOLEDs. The optical and CM images of the GSOLEDs based on a, 3M elastomer and b,
thin elastomer showed that there were as much buckling on the pixel area as on the non-pixel
area. Average buckling periodicity on the pixel area of 3M elastomer was 280 µm, whereas
buckling on the non-pixel area was 309 µm. When thin elastomer was used, the buckling
periodicity on pixel and non-pixel area was 156 µm and 121 µm, respectively. Thus, thinner

elastomer is helpful to make smaller buckling periodicity, however, we are limited by the current
 strain threshold of the thin elastomer materials for the realization of small buckling devices.”

(Page 13, Supplementary Figure S7 in the revised supplementary information)

(Page 13, line 2 ~ 10 in the revised supplementary information)

“Figure S7. Camera images and schematic illustration of the pixel area under the different
 strain on the GSOLEDs. The camera images of buckled pixel area with a, the 3M elastomer
 and b, thin elastomer. For the complete comparison, we have added the schematic illustration
 below the camera images. At first, we transferred the OLED devices on pre-strained elastomers
 (3M elastomer and thin elastomer) until their limit (100% and 30%). When we released the strain
 from the elastomer as shown in Fig. S4, the pixel areas (5 mm × 5 mm) shrunk to 3.5 mm × 3.5
 14 mm (3M elastomer) and 4.4 mm × 4.4 mm (thin elastomer) according to the initial sizes of the
 15 elastomers. Thus, the 2D strain in the GSOLEDs can be defined as percentage of the initially
 deformed pixel area (A_i) and deformed pixel area (A_d) with respect to the non-deformed pixel
 area ($A_{nd} = 25 \text{ mm}^2$) that is, strain = $[(A_{nd} - A_d) / A_i] \times 100\%$. i.e., 65 % strain = $[(25 \text{ mm}^2 - 16$
 $\text{mm}^2) / 12.25 \text{ mm}^2] \times 100\%$ at the 3M elastomer.”

**Comment 2:** Comparison with the other transparent electrodes are not enough. Although the
authors criticize many points of the other electrodes, there are no quantitative comparison.
MoO_x/Au/MoO_x can be also criticized in same way. MoO_x is water soluble. Au is very
expensive... If the author cannot demonstrate the clear advantage of MAM, the reviewer
recommends the authors to change the title and stop criticizing the other transparent electrodes.

**Authors Reply:** We appreciate your kind suggestions to improve our manuscript. We have tried
to emphasize on the merit of MoO₃/Au/MoO₃ (MAM) electrode with comparison to other
electrodes. Each electrode including MAM has some advantages and disadvantages; the parts of
MAM also have some areas to be improved, for example, MoO₃ dissolves in water and Au is too
expensive as suggested by reviewer #3. However, we have selected the MoO₃ and Au in OMO
electrode, based on the following two considerations. One is that MoO₃ possesses the deep
fermi-level (~6.7 eV), which is beneficial for hole injection from MAM electrode to NPB HTL
layer (~5.4 eV of HOMO). The other one is that Au has the excellent mechanical ductility, and it
possesses the ~5.2 eV of fermi-level, which is also helpful for hole injection into NPB.

Even though our approach to use the MAM electrode was successful and the comparison
table of advantages and disadvantages for each electrode (Supplementary Figure S2) was
proposed, we could not show clear evidence to prove its superiority over other electrodes. As per
your comments, we have altered the introduction parts avoiding criticizing other electrodes.
Among the OMOs, MAM was suggested as one of the promising candidates for the highly
conductive and transparent electrodes due to the deep work function of MoO₃ and the ductility of
Au. Additionally, we have also changed the title replacing “Oxide-Metal-Oxide Electrode” with
“MoO₃/Au/MoO₃ Electrode”.

**(Page 1, line 1 ~ 3 in the revised manuscript)**

“Highly Efficient, Heat Dissipating, Stretchable Organic Light-Emitting Diodes Based on an
MoO₃/Au/MoO₃ Electrode with Encapsulation”

**(Page 4, line 5 ~ 11 in the revised manuscript)**

“This interest has led to various studies on electrodes as a replacement of poly(3,4-
ethylenedioxythiophene):poly(styrene sulfonate) (PEDOT:PSS),⁶ and indium tin oxide (ITO)^{7,8,9},
which included a graphene-based electrode,¹⁰ a highly transparent metal-grid¹¹ and polymer with
silver nanowires (AgNWs).^{12, 13} Even though these approaches provide considerable
enhancements for flexible/stretchable electronics, there is still scope for further improvement in
terms of device stability and efficiency.^{6, 8, 9, 12, 13, 14,}”

**(Page 4, line 19 ~ Page 5, line 2 in the revised manuscript)**

“Among the OMOs, molybdenum trioxide (MoO₃)/gold (Au)/MoO₃ (MAM) has been found to
be one of the promising candidates for the highly conductive and transparent electrodes due to

the deep work function of MoO₃ and Au for hole injection, including the ductile property for
 stretchability, even though the MoO₃ dissolves in water and Au is expensive.^{15, 21, 22}

 **(Page 7, line 10 ~ 19 in the revised manuscript)**

“The bandgap alignment of the layers is shown in Fig. 1b. The MAM demonstrated a superior ϕ
 (5.25 eV) as compared to the Ag electrode (4.81 eV), due to the deeper conduction band of the
 MoO₃, analyzed using ultraviolet photoelectron spectroscopy (UPS) (Supplementary Figure S1).
 Transition metal oxides are utilized to reduce the hole injection barriers, hence, the sandwiched
 MoO₃ with Au is beneficial for better device performance.^{45, 46} Moreover, the MAM thicknesses
 were optimized using thin film optic theory and sheet resistance analysis.^{15, 16, 17} MoO₃(15
 11 nm)/Au(14 nm)/MoO₃(5 nm) demonstrated a fair balance between high transmittance of 70% ~
 12 80% and a low sheet resistance of ~ 20 Ω /sq, the best figure of merit^{47, 48} value ($2.96 \times 10^{-3} \Omega^{-1}$)
 (Supplementary Figure S1). The prepared MAM electrode showed a stable performance during
 stretching cyclic test (Supplementary Figure S2).”

 **(Page 5, Supplementary Figure S2 in the revised supplementary information)**

Transparent Electrodes	Advantages	Disadvantages	Ref
PEDOT:PSS	 High transmittance (>80%) Solution processable Well matched for Hole injection material Suitable for stretchable electronics (flexible) 	 Poor environmental stability ITO etched by acidic PEDOT:PSS High sheet resistance 	[12]
ITO	 High transmittance (>90%) Low sheet resistance (<20 Ohm/sq) Well matched for Hole injection electrode 	 Poor mechanical flexibility thus not suitable for stretchable electronics High deposition temperature (>300 °C) Increasing cost of indium 	[13]
AgNWs	 Suitable for intrinsically, geometrically stretchable electronics High transmittance (>80%) Low sheet resistance (10 to 50 Ohm/sq) Good mechanical flexibility 	 Rough surface Poor environmental stability (native oxide) Network meltdown into nanoparticles 	[14]
MAM [MoO ₃ (15nm) / Au (14nm) / MoO ₃ (5nm)]	 Low deposition temperature (<300 °C) High transmittance (>80%) Low sheet resistance (10 to 50 Ohm/sq) High environmental stability Suitable for geometrically stretchable devices Mechanical ductility High work function (~5.25 eV, MAM) 	 Not suitable for intrinsically stretchable, suitable geometrically stretchable electronics Water soluble material (MoO₃) Expensive material (Au) 	[1] [3]

17

**Comment 3:** The reviewer conducted additional experiments to test the performance of
encapsulations. Still, the importance of SiN_x and edge encapsulation is well known. In addition,
stretchable OLED is not demonstrated with the best encapsulation.

**Authors Reply:** Thank you for your kind comments. We absolutely agree that the importance of
SiN_x and edge encapsulation is well known. However, it is hard to find some papers on the
encapsulation of GSOLEDs, whereas it is easy to find some papers on only flexible OLEDs. To
our knowledge, this is the first trial to implement the encapsulated GSOLEDs. But, as we have
suggested in the supplementary information (Fig. S3), device A is stretchable, while device B, C
and D are not fully stretchable. There is a trade-off relation between encapsulation quality and
stretchability. Even though those encapsulations (device B, C and D) suggested improved
WVTR than that of device A, it needs further study to implement edge encapsulation using
stretchable sealant. However, it was the first trial for GSOLEDs, and it could be estimated for the
meaningful step of stretchable electronics. Additionally, we have transferred both encapsulation
data (with SiN_x and edge) and water immersion test from the revised manuscript to
supplementary information.

**(Page 7, line 19 ~ Page 8, line 7 in the revised manuscript)**

“The device stability was also reinforced by using thin NOA63 for encapsulation, which
fabricated a sandwiched structure with thin NOA63, induced a small bending strain and adjusted
the mechanical neutral plane (MNP) at the high YM layers.^{8, 27, 28, 29} Further, the encapsulation of
NOA63 with 3M tape, silicon nitride (SiN_x) (200 nm) thin film and NOA63 side passivation
presented a moderate water vapor transmission rate (WVTR) ($3.6 \times 10^{-2} \text{ g}\cdot\text{m}^{-2} \text{ day}^{-1}$,
Supplementary Figure S3) as compared with that of the recently introduced 4–6 layers of
graphene-encapsulated OLEDs ($1.78 \times 10^{-2} \text{ g}\cdot\text{m}^{-2} \text{ day}^{-1}$).^{40, 41} Nevertheless, there is a trade-off
between encapsulation quality and stretchability. However, even though flexible OLEDs have
been reported with WVTR value,^{40, 41} stretchable OLEDs with encapsulation have been reported
rarely by now. Therefore, further study is needed to implement edge encapsulation with a
stretchable sealant for the industrial requirement and practical wearable gadgets based on the
GS- or ISOLEDs.”

**(Page 8, line 3 ~ 6 in the revised supplementary information)**

“Even though the encapsulations (device B, C and D) suggested improved WVTR than that of
device A, those devices were not fully stretchable. There is a trade-off relation between
encapsulation quality and stretchability. Therefore, it needs further study to implement edge
encapsulation using stretchable sealant for GSOLEDs.”

**(Page 6, Supplementary Figure S3 in the revised supplementary information)**

Encapsulation	Device A	Device B	Device C	Device D
WVTR (gm ⁻² day ⁻¹)	4 × 10 ⁰	2.0 × 10 ⁻¹	3.6 × 10 ⁻²	1.8 × 10 ⁻²

$$\begin{aligned}
 WVTR &= \rho(CaO) \times \frac{m(H_2O)}{m(CaO)} \times \frac{D(CaO)}{t} \quad [g/m^2 day] \\
 &= 3.35 \times 10^6 g/m^3 \times \frac{18.01528g/mol}{56.077g/mol} \times D(CaO)/t \quad [g/m^2 day] \\
 &= 1.07622 \times 10^6 g/m^3 \times D(CaO)/t \quad [g/m^2 day] \\
 &\Rightarrow 1.07622 \times 10^6 g/m^3 \times \left(\frac{T(Final) - T(Initial)}{100 - T(Initial)} \times 500 \times 10^{-10} m \right) / \left(Time(min) \times \frac{1h}{60min} \times \frac{1day}{24h} \right)
 \end{aligned}$$

* ρ: Density, m: Molar concentration, D: Thickness, t: Time, T: Transmittance

Comment 4: In page 34 of the rebuttal letter, the authors summarize their work. Here the reviewer shows their opinion on each point.

i) The device lifetime of GSOLEDs in various condition, such as encapsulation method,

water immersion and strain condition.

As commented in this comment 3, it is difficult to find the novelty.

**Authors Reply:** Thank you for kind suggestions. We were inspired from the previously
published paper [ACS Appl. Mater. Inter. 2016, 8, 31166–31171, Ref 28]. However, we would
like to emphasize that our work has several novelties as compared to other works. Following
your suggestions in comment 3, we have transferred both encapsulation data (with SiN_x and edge
encapsulation) and water immersion test from the revised manuscript to supplementary
information.

However, we have conducted various stability tests such as twist mode (0°, 90° and 180°) and
LT50 measurement under the strain (0%, 30%, 65% and 100%). Moreover, we have performed
the mechanical simulation of 2-D random buckling for verifying the applied strain on the pixel in
GSOLEDs platform. Although our encapsulation was not that perfect, as mentioned in comment
#3, we have suggested the possibility of encapsulation improvement on GSOLEDs. So, we are
planning to get more research for side encapsulation and passivation for GS or ISOLEDs using
stretchable sealant. In addition to those explored properties mentioned above, some other
novelties have also been explained in the manuscript including a negligible roll-off in the devices,
no EL shift between buckled and planar modes, higher outcoupling in stretchable OLEDs and so
on.

**(Page 36, Figure 4 in the revised manuscript)**

ii) The GSOLEDs based on the MAM with the invariant color-coordinate.
 The invariant color coordinate happens only when nontransparent electrodes are used. It is not a
 general issue for all the stretchable OLEDs. In addition, the importance of the use of MAM is not
 demonstrated as the reviewer mentioned in Comment 2.

**Authors Reply:** Thank you for valuable comments. We agree with your opinion that the
 variation in color coordinates is not a common problem with OMO electrode, comparing to
 semi-transparent electrode, such as Ag 20 nm. However, we would like to emphasize that our
 devices do not change their color coordinate even under severe strain (0%, 30%, 65% and 100%)
 in the GSOLEDs platform using MAM electrode. To our experience, this is the unique trial to
 demonstrate the GSOLEDs using MAM, while the electrode remains intact by applying
 mechanical strain.

For the wearable electronics, the invariant color in GS- or ISOLEDs is quite important up to -
 $70^\circ \sim 70^\circ$ viewing angle and $0 \sim 100\%$ strain simultaneously. As discussed on comment #2, we
 have suggested that MAM electrode can be a potential candidate for the stretchable devices with
 high durability.

(Page 38, Figure 6 in the revised manuscript)

 iii) The relation in GSOLEDs between substrate thickness and heat dissipation through the
 TREL and suggestion of the exact mechanism as shown in Fig. 6.

Heat dissipation by thin substrate maybe a major novelty of this paper, but the current
 manuscript contains too many not-so-novel parts. The reviewer recommends to cut down these
 parts and revise the manuscript dramatically. In addition, the authors should compare the heat-
 dissipation performance with the other approaches.

**Authors Reply:** Thank you for your kind comments. As you pointed out, we have modified the
 manuscript by removing the unnecessary/additional parts or transferring it to the supplementary
 information. According to your suggestion, we have compared the heat-dissipation performance
 of our approach with other approaches (Supplementary Information Table S2). Among the
 various tactics, our approach stands out to be effective way to dissipate heat from OLEDs.

In general, thermal conductivity of NOA63 ($0.2 \text{ W/m}^{-1} \text{ K}^{-1}$) is lower than that of glass or
 other substrates. However, NOA63 ($9.2 \mu\text{m}$) being very thin, its thermal conductance is very
 high compared to thicker glass substrate. In that way, our heat dissipation works effectively with
 thin elastomer and SiO_2 NPs. Additionally, we have concluded from the TREL result that the
 incorporation of SiO_2 NPs with NOA63 improves the heat dissipation of the devices.

For enhancing the readability, we have reduced the total numbers of main figures from 8 to 6
 and of Supplementary Figures from 26 to 20. We have rearranged the figures and associated
 contents for focusing on our novelties, which would be discussed in comment 5.

(Page 37, Figure 5 in the revised manuscript)

1

2 (Page 33, Supplementary Table S2 in the revised supplementary information)

Technique (Methods)	Temperature deviation compared to control device (ΔT , °C)	Efficiency improvement compared to control device (Δ cd/A, Δ)	Reference
Thin film encapsulation (Liquid getter)	7.7 °C @ 8 V	Not reported	[29, 37]
Heat sink (Cu layer)	8.5 °C and 11.5 °C @ 6 V and 9 V	Not reported	[38]
Heat sink (Heat transfer fluid)	Not measured	70% enhancement	[39]
Heat sink (Conducting substrate)	24 °C (for SUS) 43.7 °C (for silicon substrate)	3.4 % @ 8V (for SUS and silicon substrate)	[40]
Thin film encapsulation (Inorganic / organic / metal)	2.23 °C (with TFE) 4.51 °C (with MET-TFE)	Not reported	[41]
Heat dissipation (Thin substrate + SiO ₂ NPs)	11.4 °C @ 10,000 nits	40% enhancement @ 10,000 nits	Current Work

(Page 14, line 3 ~ 4 in the revised manuscript)

“The heat-dissipation performance facilitated by thin elastomer and SiO₂ NPs was found to be
effective as compared to other techniques (Supplementary Table S2).”

iv) The demonstration of the stretchable and heat dissipating device with thin elastomer (100
μ m), instead of 3M elastomer (1,000 μ m) simultaneously as shown in Fig. 4a, b. (To our
knowledge, it has not been reported before)13 It may not be reported, but the definition of the stretchability of this device is doubtful as written
in my comment 1.15 **Authors Reply:** Thank you for valuable comments. We have suggested 3M elastomer (1,000 μ m)
devices can be stretchable until 100% strain. However, thin elastomer (100 μ m) devices can be
stretchable only until 30% because it is too brittle to sustain the strain beyond 30%. Buckling of
thin elastomer is much smaller than that of 3M elastomer due to the difference in pre-strain. If a
more robust elastomer capable of forming a thin film with over 30% strain is realized, we are
positive that our proposed method is capable of demonstrating even better results.

To make the stretchability of our devices understandable to the readers, we have fabricated new devices with 3M elastomer (1,000 μm) and thin elastomer (100 μm). As we discussed in comment #1, we have measured the exact emitting area with confocal and camera images. Also, we have added the relation of stretchability and buckling in the revised manuscript accordingly.

(Page 8, line 23 ~ Page 9, line 2 in the revised manuscript)

“The buckling of thin elastomer showed smaller buckling compared to thick 3M elastomer. Additionally, we have defined the strain based on light emitting area (pixel area) with both types of elastomers (Supplementary Figure S7).”

(Page 13, Supplementary Figure S7 in the revised supplementary information)

v) The application of the various encapsulation efficiently for GSOLED with enhanced WVTR (detailed explanation in comment 3)

This investigation is not new as described in comment 3. Much better water stability of flexible
OLEDs has been shown elsewhere.

**Authors Reply:** Thank you for kind suggestions. As you already mentioned on comments 3,
WVTR with SiN_x is well known. Even though flexible OLEDs have been reported with WVTR
value, stretchable OLEDs with encapsulation have been reported rarely by now. Here, we have
demonstrated WVTR of stretchable OLEDs with encapsulation. However, as you mentioned in
previous revision, WVTR of device A is not much impressive. Implementing encapsulation on
GSOLEDs without affecting the stretchability requires further study. We are planning to
explore edge encapsulation for GS- or ISOLEDs using stretchable sealant in our next project.
Please verify our detailed responses in comment 3

vi) The SiO₂ NPs are beneficial for both optical out-coupling and the heat dissipation
mechanism, concurrently.

As pointed out by the other reviewer, this point is not new. as shown in the following paper, high
optical out-coupling and higher EQE is already demonstrated just by reducing the substrate
thickness. <https://doi.org/10.1016/j.orgel.2019.03.040>

**Authors Reply:** Thank you for kind suggestions. The efficiency is impressive with just reducing
the parylene substrate thickness (with changing the *n* value compared to glass). We have also
tried to get thinner substrate, but we could not achieve that due to the constraint of our equipment
(~5000 RPM spin coater limit). Because of that, as a plan B, we have fabricated the device with
thin elastomer. Even though the elastomer is not that much thin, the device with thin elastomer
did not show any roll-off.

We agree that addition of SiO₂ NPs to improve the light out coupling is not new. However,
we have found that porous scaffold design of SiO₂ NPs is also helpful for heat dissipation. While
the thickness of NOA63 with SiO₂ NPs is almost twice than that of NOA63 without SiO₂, metal
oxide NPs help to dissipate the heat efficiently. Additionally, we have suggested and investigated
the influence of SiO₂ NPs on the heat dissipation mechanism with TREL as shown in Fig. 5 in
the revised manuscript.

**(Page 37, Figure 5 in the revised manuscript)**

Comment 5: Current manuscript has no focus in the manuscript and is too long. The manuscript is very tough to read even for reviewers. Nature communications attract lots of attention from researchers in various field. The reviewer strongly recommend to revise the manuscript dramatically.

Authors Reply: Thank you for your valuable comments to improve the quality of our work. We have revised the manuscript as per your comments and suggestions. For enhancing the readability, we have reduced the total numbers of main figures from 8 to 6 and of Supplementary Figures from 26 to 20. We have rearranged the figures and associated contents for focusing on our novelties.

Old (1 st revision)	Revised (2 nd revision)	Contents
Fig. 4a, b	Fig. 3b, e	Device performance of the thin elastomer based GSOLED devices
Fig. 5	Fig. 4a-c	Device performance on twisting
Fig. 4e and S6f	Fig. 4d-f	LT50 of the devices at various 2D stretch percentages (0-100%) and the device performance stability of the cyclic test

Fig. 6	Fig. 5	Device performance of the MoO ₃ /Au/MoO ₃ (MAM) based GSOLED devices by heat dissipation mechanism
Fig. 7	Fig. 6	Mechanism of optical light scattering along with the EL and color coordinate analysis
Fig. 8	Fig. 4g and S13i	Mechanical simulation analysis of the various mechanisms in the GSOLED
Supplementary Figure S6	Removed	Device performance of GSOLEDs with Ag and MAM electrodes at various stretch percentages (0-100%)
Supplementary Figure S7	Removed	AFM and UHR FE-SEM analysis of NOA63 with and without NPs along electrodes. The contact angle (CA) of distilled ionized water (DIW) on various substrates
Supplementary Figure S8	Removed	Hole-only-device (HOD) analysis with the Ag, MoO ₃ /Au (MA) and MoO ₃ /Au/MoO ₃ (MAM) electrodes.
Supplementary Figure S9	Removed	Device performance optimization by varying the hole transport layer (HTL) thickness
Supplementary Figure S10	Removed	Device performance comparison of OLED devices with various-sized (50, 100, 500 and 700 nm) SiO ₂ NPs embedded in the NOA63 substrate
Supplementary Figure S11	Removed	Device performance of the OLED devices with different substrates including glass, NOA63 with and without SiO ₂ NPs
Supplementary Figure S12	Removed	The device performances with/without NPs are compared, with both thin NOA63 with/without 3M elastomer with (a, b) isotropic and (c, d) horizontal emitters
Supplementary Figure S13a,b	Removed	The performance of the heat dissipating devices with (a, b) isotropic and different thicknesses of elastomers with and without NPs
Supplementary Table S2 ~ S6	Removed	Device performance (CE, EQE and PE) of the GSOLED devices on NOA63 with and without NPs
Supplementary Figure S26	Removed	Various modes photos for the GSOLEDs in working and testing modes

We have highlighted the revised sentences in **blue color**. With the incorporated changes, we
hope the manuscript is now acceptable for publication in the **Nature Communications**. Should
you have any further questions or comments, please do not hesitate to contact me.

Sincerely,

Seung Yoon Ryu

Division of Display & Semiconductor Physics,

College of Science and Technology,

Korea University (Sejong Campus),

(30019) 2511, Sejong-ro, Sejong,

Republic of Korea

REVIEWERS' COMMENTS

Reviewer #3 (Remarks to the Author):

The author revised the text and the manuscript became easy to follow. However, the manuscript does not show the novelty enough for Nature Communications. Therefore, the reviewer recommends the rejection. The followings are the details.

1. Stretchability

As authors admitted in the reply to my previous comment #1, stretchability mainly comes from very large flexing of the substrate. This is clearly observed in Figure S5. If this can be called stretchable, all the flexible OLEDs can be called stretchable. The authors should compare the size of active area by laminating the device on flat surface.

2. Advantage of OMO

As the author admitted in the reply to my previous comment #2, the authors do not demonstrate the advantage of OMO over other transparent electrodes. As shown in other papers using transparent electrode, OLEDs or PLEDs show no change in color coordinate. In addition, all the references in Fig. S2 does not support the advantages and disadvantages in the table.

3. Encapsulation of GSOLEDs

Although the reviewer evaluates the authors' effort, the results shown in Fig. S3 has no new findings. In addition, encapsulation of GSOLEDs have been previously discussed in [10.1126/sciadv.1501856](https://doi.org/10.1126/sciadv.1501856).

4. Heat dissipation

This method might be new but shows no clear advantage compared to others. Table S2 is not so clear to claim "The heat-dissipation performance facilitated by thin elastomer and SiO₂ NPs was found to be effective as compared to other techniques" as texted in Page 14, Line 3.

5. EQE improvement

The authors admitted just reducing the thickness is enough for improving EQE and GSOLEDs has been demonstrated using such a thin plastic substrates.

6. In Page 9, Line 14, what do you mean by "Apple Silicon"?

Reviewer(s)' Comments to Author:

REVIEWER COMMENTS

Reviewer #3

Recommendation: Rejection

The author revised the text and the manuscript became easy to follow. However, the manuscript does not show the novelty enough for Nature Communications. Therefore, the reviewer recommends the rejection. The followings are the details.

Comment 1: Stretchability

As authors admitted in the reply to my previous comment #1, stretchability mainly comes from very large flexing of the substrate. This is clearly observed in Supplementary Fig. S5. If this can be called stretchable, all the flexible OLEDs can be called stretchable. The authors should compare the size of active area by laminating the device on flat surface.

Authors Reply: Thank you for your comments. We understood the reviewer's concern. However, the concept of flexible OLEDs and stretchable OLEDs are different and should be well understood. Stretchable OLEDs have changeable form factor, while flexible OLEDs have fixed one. In other words, the pixel area including overall buckling in the stretchable devices changes, whereas the pixel area does not change in flexible devices. In the Fig S5, it is clearly shown that there is sufficient buckling inside the pixels that allows the OLEDs to be stretchable. Regarding the comparison of the periodicity of the buckling, we have already compared buckling formation inside and outside of pixel area. We have found that there was no big difference between the buckling of the pixel and non-pixel area. Additionally, we have already demonstrated that much smaller buckling formation with thin elastomer (Supplementary Fig. S5). This buckling-based geometrically stretchable OLED demonstrated mechanical stability in various operation modes (1-dimensional, 2-dimensional stretching and twisting including cycle test).

(Page 8, line 9 ~ 14 in the revised manuscript)

“The fabricated wavy buckles were observed by optical microscope (OM) and confocal microscope (CM) on the NOA63, with and without NPs. We have also measured the pixel area and non-pixel area with CM for verifying buckling on the different parts of the devices (Supplementary Fig. S5 and S6). The buckling of thin elastomer showed smaller buckling compared to thick 3M elastomer. Additionally, we have defined the strain based on light emitting area (pixel area) with both types of elastomers (Supplementary Fig. S7).”

(Page 10, Supplementary Fig. 5 in the revised supplementary information)

(Page 10, line 2 ~ 8 in the revised supplementary information)

“Figure S5. Camera and confocal microscopy (CM) images of pixel and non-pixel area in the GSOLEDs. The optical and CM images of the GSOLEDs based on a, 3M elastomer and b, thin elastomer showed that there were as much buckling on the pixel area as on the non-pixel area. Average buckling periodicity on the pixel area of 3M elastomer was 280 μm , whereas buckling on the non-pixel area was 309 μm . When thin elastomer was used, the buckling periodicity on pixel and non-pixel area was 156 μm and 121 μm , respectively. Thus, thinner elastomer is helpful to make smaller buckling periodicity, however, we are limited by the current strain threshold of the thin elastomer materials for the realization of small buckling devices.”

Comment 2: Advantage of OMO

As the author admitted in the reply to my previous comment #2, the authors do not demonstrate the advantage of OMO over other transparent electrodes. As shown in other papers using transparent electrode, OLEDs or PLEDs show no change in color coordinate. In addition, all the references in Fig. S2 does not support the advantages and disadvantages in the table.

Authors Reply: Thank you for your kind comments. We have already tried to demonstrate the advantage and disadvantage of $\text{MoO}_3/\text{Au}/\text{MoO}_3$ (MAM) electrode with other electrode candidates (Supplementary Fig. S2). Overall, there are three major advantages that we have chosen for the MAM electrode. Firstly, as shown in ultraviolet photoelectron spectroscopy (UPS) (Supplementary Fig. S1b) analysis, MAM showed a suitable fermi level ($\sim 5.25\text{eV}$) for hole injection into NPB ($\sim 5.4\text{eV}$). Secondly, inner Au layer has the noteworthy mechanical ductility, which is helpful for geometrically stretchable OLEDs. Finally, MAM electrode is suitable for maintaining color purity optically under various mechanical deformation. In addition, the structure of the MAM electrode has optimized transmittance and resistance values under the analysis of figure of merit (FOM) (Supplementary Fig. S1). Also, the references of other candidate electrode are sufficient for understanding the advantages and disadvantages in the table (Supplementary Fig. S2).

(Page 3, line22 ~ Page 5, line 14 in the revised manuscript)

“This interest has led to various studies on electrodes as a replacement of poly(3,4-ethylenedioxythiophene):poly(styrene sulfonate) (PEDOT:PSS),⁶ and indium tin oxide (ITO)^{7, 8, 9}, which included a graphene-based electrode,¹⁰ a highly transparent metal-grid¹¹ and polymer with silver nanowires (AgNWs).^{12, 13} Even though these approaches provide considerable enhancements for flexible/stretchable electronics, there is still scope for further improvement in terms of device stability and efficiency.^{6, 8, 9, 12, 13, 14} Transparent electrode based on dielectric/metal/dielectric and an oxide-metal-oxide (OMO), where a thin metal layer is sandwiched between two dielectric or oxide layers, is similarly a promising candidate. That can

be tuned to produce highly conductive and transparent multilayer electrode exploiting the conductivity of the metal, the optical interference within the multilayers and the surface plasmonic effect at the metal/oxide interface.^{15, 16, 17, 18, 19} In terms of flexibility and stretchability, a thin and ductile metal layer has the role of resisting fracture since it acts as the crack stopper or retardation on an elastomeric substrate.¹⁹ Therefore, a thin OMO electrode could be proposed for stretchable electronics because of its better hole injection and optical transmittance, despite the thick oxide material being known for brittleness under high strains.²⁰ Among the OMOs, molybdenum trioxide (MoO₃)/gold (Au)/MoO₃ (MAM) has been found to be one of the promising candidates for the highly conductive and transparent electrodes due to the deep work function of MoO₃ and Au for hole injection, including the ductile property for stretchability, even though the MoO₃ dissolves in water and Au is expensive.^{15, 21, 22} ”

(Page 6, line 21 ~ Page 7, line 9 in the revised manuscript)

“The bandgap alignment of the layers is shown in Fig. 1b. The MAM demonstrated a superior ϕ (5.25 eV) as compared to the Ag electrode (4.81 eV), due to the deeper conduction band of the MoO₃, analyzed using ultraviolet photoelectron spectroscopy (UPS) (Supplementary Fig. S1). Transition metal oxides are utilized to reduce the hole injection barriers, hence, the sandwiched MoO₃ with Au is beneficial for better device performance.^{45, 46} Moreover, the MAM thicknesses were optimized using thin film optic theory and sheet resistance analysis.^{15, 16, 17} MoO₃(15 nm)/Au(14 nm)/MoO₃(5 nm) demonstrated a fair balance between high transmittance of 70% ~ 80% and a low sheet resistance of ~ 20 Ω /sq, the best figure of merit^{47, 48} value ($2.96 \times 10^{-3} \Omega^{-1}$) (Supplementary Fig. S1). The prepared MAM electrode showed a stable performance during stretching cyclic test (Supplementary Fig. S2 and Supplementary video 1).”

(Page 5, Supplementary Fig. 2 in the revised supplementary information)

Transparent Electrodes	Advantages	Disadvantages	Ref
PEDOT:PSS	 High transmittance (>80%) Solution processable Well matched for Hole injection material Suitable for stretchable electronics (flexible) 	 Poor environmental stability ITO etched by acidic PEDOT:PSS High sheet resistance 	[12]
ITO	 High transmittance (>90%) Low sheet resistance (<20 Ohm/sq) Well matched for Hole injection electrode 	 Poor mechanical flexibility thus not suitable for stretchable electronics High deposition temperature (>300 °C) Increasing cost of indium 	[13]
AgNWs	 Suitable for intrinsically, geometrically stretchable electronics High transmittance (>80%) Low sheet resistance (10 to 50 Ohm/sq) Good mechanical flexibility 	 Rough surface Poor environmental stability (native oxide) Network meltdown into nanoparticles 	[14]
MAM [MoO ₃ (15nm) / Au (14nm) / MoO ₃ (5nm)]	 Low deposition temperature (<300 °C) High transmittance (>80%) Low sheet resistance (10 to 50 Ohm/sq) High environmental stability Suitable for geometrically stretchable devices Mechanical ductility High work function (~5.25 eV, MAM) 	 Not suitable for intrinsically stretchable, suitable geometrically stretchable electronics Water soluble material (MoO₃) Expensive material (Au) 	[1] [3]

Comment 3: Encapsulation of GSOLEDs

Although the reviewer evaluates the authors' effort, the results shown in Fig. S3 has no new findings. In addition, encapsulation of GSOLEDs have been previously discussed in 10.1126/sciadv.1501856.

Authors Reply: Thank you for your indication. We agree that the encapsulation is not new in GSOLEDs. As your comment, we have tried to revise the manuscript accordingly. However, we have tried to verify the encapsulation reliability through water vapor transmittance ratio (WVTR) analysis for various encapsulation methods. We have also conducted water immersion test and provided the devices lifetime. However, we think that the additional study on the stretchable sealant for improving WVTR value is needed for stretchable electronics. We believe that our research can be a useful approach for studying encapsulation on stretchable devices. We have modified our manuscript acknowledging passivation for flexible and stretchable OLEDs have been used in the previously published paper [Yokota T, et al. Ultraflexible organic photonic skin. Sci Adv 2, e1501856 (2016), Ref 42].

(Page 5, line 21 ~ Page 6, line 2 in the revised manuscript)

“Although, the passivation for flexible and stretchable OLEDs have been demonstrated previously,^{40, 41, 42} various approaches of encapsulation for water-proofing, and mechanical analysis of the strain have not been discussed.”

(Page 7, line 9 ~ 18 in the revised manuscript)

“The device stability was also reinforced by using thin NOA63 for encapsulation, which fabricated a sandwiched structure with thin NOA63, induced a small bending strain and adjusted the mechanical neutral plane (MNP) at the high YM layers.^{8, 27, 28, 29} Further, the encapsulation of NOA63 with 3M tape, silicon nitride (SiN_x) (200 nm) thin film and NOA63 side passivation presented a moderate water vapor transmission rate (WVTR) ($3.6 \times 10^{-2} \text{ g}\cdot\text{m}^{-2} \text{ day}^{-1}$, Supplementary Fig. S3) as compared with that of the recently introduced 4–6 layers of graphene-encapsulated OLEDs ($1.78 \times 10^{-2} \text{ g}\cdot\text{m}^{-2} \text{ day}^{-1}$).^{40, 41} Nevertheless, there is a trade-off between encapsulation quality and stretchability. However, even though flexible OLEDs have been reported with WVTR value.^{40, 41} Therefore, further study is needed to implement edge encapsulation with a stretchable sealant for the industrial requirement and practical wearable gadgets based on the GS- or ISOLEDs.”

(Page 7, line 23 ~ Page 8, line 2 in the revised supplementary information)

“Even though the encapsulations (device B, C and D) suggested improved WVTR than that of device A, those devices were not fully stretchable. There is a trade-off relation between encapsulation quality and stretchability. Therefore, it needs further study to implement edge encapsulation using stretchable sealant for GSOLEDs.”

(Page 6, Supplementary Fig. S3 in the revised supplementary information)

Comment 4: Heat dissipation

This method might be new but shows no clear advantage compared to others. Table S2 is not clear to claim "The heat-dissipation performance facilitated by thin elastomer and SiO_2 NPs was found to be effective as compared to other techniques" as texted in Page 14, Line 3.

Authors Reply: Thank you for your kind comments. We have tried our best to show the heat-dissipation performance of our device compared with other published works in Table S2, which clearly indicates the advantages of thin elastomer and SiO_2 NPs in heat-dissipation. Especially, we have concluded the heat dissipation properties from the TREL result. The incorporation of

SiO₂ NPs with NOA63 enhanced the heat dissipation due to the porous scaffold design. Even though the heat conductivity of NOA63 is lower than that of conventional heat transfer materials (e.g., glass substrate), the devices with thinner substrate showed efficient heat dissipation property due to the higher thermal conductance of the substrate.

(Page 33, Supplementary Table S2 in the revised supplementary information)

Technique ^⓪ (Methods) ^⓪	Temperature deviation ^⓪ compared to control device ^⓪ (ΔT , °C) ^⓪	Efficiency improvement ^⓪ compared to control device ^⓪ (Δ cd/A, Δ) ^⓪	Reference ^⓪
Thin film encapsulation ^⓪ (Liquid getter) ^⓪	7.7 °C @ 8 V ^⓪	Not reported ^⓪	[29, 37] ^⓪
Heat sink ^⓪ (Cu layer) ^⓪	8.5 °C and 11.5 °C ^⓪ @ 6 V and 9 V ^⓪	Not reported ^⓪	[38] ^⓪
Heat sink ^⓪ (Heat transfer fluid) ^⓪	Not measured ^⓪	70% enhancement ^⓪	[39] ^⓪
Heat sink ^⓪ (Conducting substrate) ^⓪	24 °C (for SUS) ^⓪ 43.7 °C (for silicon substrate) ^⓪	3.4 % @ 8V ^⓪ (for SUS and silicon substrate) ^⓪	[40] ^⓪
Thin film encapsulation ^⓪ (Inorganic / organic / metal) ^⓪	2.23 °C (with TFE) ^⓪ 4.51 °C (with MET-TFE) ^⓪	Not reported ^⓪	[41] ^⓪
Heat dissipation ^⓪ (Thin substrate + SiO ₂ NPs) ^⓪	11.4 °C @ 10,000 nits ^⓪	40% enhancement ^⓪ @ 10,000 nits ^⓪	Current ^⓪ Work ^⓪

(Page 36, Fig. 5e in the revised manuscript and Page 22, Supplementary Fig. S15b in the revised supplementary information)

(Page 12, line 14 ~ Page 13, line 21 in the revised manuscript)

“The generated and accumulated heat in thick substrate devices provided thermally activated energy to triplet excitons, which boosted longer exciton lifetimes, TTA, and TPA, finally degrading the device efficiency at high exciton density.^{30, 31, 32, 33, 34} However, heat dissipated in thin NOA63 substrate devices efficiently avoided exciton annihilation and sustained or improved device efficiency (Fig. 5a). This was consistent with the time-resolved EL (TREL) measurement, which reveals that the exciton lifetimes from thick glass and NOA63 devices were longer with a monotonic decay than those of thin NOA63 devices with double-exponential features (Fig. 5e and Supplementary Fig. S15). The TREL results showed overlapping overall, except that of glass based device and the detailed comparison could be confirmed (Supplementary Fig. S15b). On the other hand, although the thickness of the NOA63 containing NPs (NPs 4.5 wt%, 16.3 μm thickness) was twice that of the film without NPs (NOA63 9.2 μm), their identical TREL results (Supplementary Fig. S15c) indicate that the effect of thicker film in dissipating heat was minimized by the presence of NPs. It was reported that Aluminum Oxide (Al_2O_3) NPs dissipated heat efficiently from perovskite solar cells due to their highly, thermally conductive ($20 \sim 30 \text{ W m}^{-1} \text{ K}^{-1}$) metal oxide NPs and porous scaffold design (the schematic illustration of Fig. 6e inset).³⁵ Since the thermal conductivity of SiO_2 NPs ($1.3 \text{ W m}^{-1} \text{ K}^{-1}$)⁶¹ is much higher than NOA63 ($0.2 \text{ W m}^{-1} \text{ K}^{-1}$), it can be inferred that SiO_2 NPs also facilitate heat dissipation from the device in a similar manner. In addition, the TREL of the device with thin elastomer (100 μm) appeared between those with 70 μm and 210 μm NOA63 based devices without elastomer, which means that the thin elastomer efficiently dissipated the heat from the inside of the device. Consequently, the total thickness of the substrate (polymer film and elastomer) was dominant in the heat dissipation mechanism. The heat-dissipation performance facilitated by thin elastomer and SiO_2 NPs was found to be effective as compared to other techniques (Supplementary Table S2). The efficient heat dissipation process might disturb any free-carrier (excitons) interactions,

such as TTA, because exciton recombination occurring in lower energy states (trapping sites) becomes dominant in an environment with lower thermal energy.^{32, 33, 34} We believe that this observation is evidence for supporting the heat dissipation mechanism for thin NOA63 devices, where the faster heat exit, reduced the TTA process.^{32, 33, 34} Therefore, a thinner elastomer is preferable to avoid the heat accumulation and exciton annihilation can be further suppressed. That is why the phosphorescent OLEDs with thick glass substrate show the efficiency roll off at high exciton densities as compared to the NOA63 devices, which indicates that the lower substrate thickness is preferable to avoid it.”

(Page 19, Fig S12 in the revised supplementary information)

(Page 11, line 14 ~ 21 in the revised manuscript)

“The heat conductivity is related to the thickness normalization, which was $0.2 \text{ W m}^{-1} \text{ K}^{-1}$ and $1.05 \text{ W m}^{-1} \text{ K}^{-1}$ for thin NOA63 and thick glass substrates, respectively. This indicates that the glass substrate itself intrinsically transfers more heat than a NOA63 film.^{30, 31} However, the heat conductance (along the direction normal to the substrate) is dependent on the relative substrate thickness, which for thin NOA63 and thick glass substrate are 543.38 and 37.5 [W K^{-1}]. This means that a thin NOA63 substrate would transfer heat more efficiently to ambient air than a thick glass substrate^{30, 31} (Supplementary Fig. S12).”

Comment 5: EQE improvement

The authors admitted just reducing the thickness is enough for improving EQE and GSOLEDs has been demonstrated using such a thin plastic substrates.

Authors Reply: Thank you for your suggestion. This is how we explained the benefit of using thin elastomer in dissipating heat from the GSOLEDs. Also, according to the reviewer’s previous comments, we have tried to fabricate GSOLEDs with thinner elastomers and updated our manuscript with the findings. Particularly, the investigation of various thickness substrate devices has compared under device performance and TREL result, which validates our conclusion. The NOA63 substrate devices showed shorter monotonic decay than thick glass substrate devices, which means heat dissipated efficiently at the high exciton density from device. In near future, we are considering to fabricate the thinner device for verifying your suggestion.

(Page 36, Fig. 5 in the revised manuscript)

(Page 12, line 16 ~ 21 in the revised manuscript)

“However, heat dissipated in thin NOA63 substrate devices efficiently avoided exciton annihilation and sustained or improved device efficiency (Fig. 5a). This was consistent with the time-resolved EL (TREL) measurement, which reveals that the exciton lifetimes from thick glass

and NOA63 devices were longer with a monotonic decay than those of thin NOA63 devices with double-exponential features (Fig. 5e and Supplementary Fig. S15).”

Comment 6: In Page 9, Line 14, what do you mean by "Apple Silicon"?

Authors Reply: We are sorry for making confusion. The thin elastomer that we used was from DOW Corning but processed by Apple Silicone, Korea (www.applesilicone.com). We have updated the manuscript accordingly.

(Page 9, line 2 ~ 4 in the revised manuscript)

“Additionally, we have replaced the thick 3M elastomer with another thin elastomer (100 μm , DOW Corning HTV processed by Apple Silicone, Korea) and the detailed device physics is discussed later.”

(Page 18, line 1 ~ 3 in the revised manuscript)

“**Buckling mechanism with a biaxially pre-strained elastomeric substrate:** A 3M VHB elastomer (3M, USA) and a thin elastomer (DOW Corning HTV processed by Apple Silicone, Korea) were utilized as the receiving substrate (25 cm^2).”

We have highlighted the revised sentences in **blue color**. With the incorporated changes, we hope the manuscript is now acceptable for publication in the **Nature Communications**. Should you have any further questions or comments, please do not hesitate to contact me.

Sincerely,
Seung Yoon Ryu
Division of Display & Semiconductor Physics,
College of Science and Technology,
Korea University (Sejong Campus),
(30019) 2511, Sejong-ro, Sejong,
Republic of Korea